# Cowpox Viruses: A Zoo Full of Viral Diversity and Lurking Threats

**DOI:** 10.3390/biom13020325

**Published:** 2023-02-08

**Authors:** Ryan C. Bruneau, Loubna Tazi, Stefan Rothenburg

**Affiliations:** Department of Medial Microbiology and Immunology, School of Medicine, University of California Davis, Davis, CA 95616, USA

**Keywords:** poxviruses, cowpox virus, vaccinia virus, zoonosis

## Abstract

Cowpox viruses (CPXVs) exhibit the broadest known host range among the *Poxviridae* family and have caused lethal outbreaks in various zoo animals and pets across 12 Eurasian countries, as well as an increasing number of human cases. Herein, we review the history of how the cowpox name has evolved since the 1700s up to modern times. Despite early documentation of the different properties of CPXV isolates, only modern genetic analyses and phylogenies have revealed the existence of multiple *Orthopoxvirus* species that are currently constrained under the CPXV designation. We further chronicle modern outbreaks in zoos, domesticated animals, and humans, and describe animal models of experimental CPXV infections and how these can help shaping CPXV species distinctions. We also describe the pathogenesis of modern CPXV infections in animals and humans, the geographic range of CPXVs, and discuss CPXV–host interactions at the molecular level and their effects on pathogenicity and host range. Finally, we discuss the potential threat of these viruses and the future of CPXV research to provide a comprehensive review of CPXVs.

## 1. Introduction

Members of the *Orthopoxvirus* genus are large double-stranded DNA (dsDNA) viruses that replicate exclusively in the host cell’s cytoplasm [1]. They are well known and studied due to their roles in shaping human history and health. Perhaps the most notorious orthopoxvirus is the variola virus (VARV), the causative agent of smallpox, which is thought to have caused epidemics in humans for thousands of years and is estimated to have caused 300 million deaths in the 20th century alone [2]. VARV was the first eradicated human disease by vaccination with vaccinia virus (VACV), which causes much milder infections in humans and confers immune cross-protection against other orthopoxvirus infections [3,4,5]. Recent outbreaks of feral VACV in Brazil [6,7] and India [8], among other countries, as well as the current global mpox (formerly known as monkeypox) outbreak, which is caused by the monkeypox virus and has infected more than over 80,000 people, illustrates that orthopoxviruses still pose major threats to human health [9]. The triumph of VACV vaccination over VARV has historically overshadowed an orthopoxvirus that played a major role in the development of the process: the cowpox virus (CPXV) (species designation: *Cowpox virus*). The CPXV name is derived from “cow-pox”, a viral disease famous for its role in the popular narrative of the discovery of vaccination. Material from cow-pox patients contained the first known disease agent that was used to induce immunity against a distinct, yet related, infectious agent leading to the development of vaccination as popularized by Edward Jenner. Afterwards, the cow-pox agent was also called vaccinia and the process of vaccination spread across the world [10,11]. This discovery has saved countless human lives and revolutionized biology, giving rise to the scientific disciplines of immunology and virology. While cow-pox is acknowledged as a precursor to VACV, popular accounts often lack discussions of modern understandings of the distinction between ancient cow-pox, both ancient and modern VACV, and modern CPXV [12,13]. This misunderstanding is not helped by modern research that demonstrates that the CPXV name encompasses genetically distinct dsDNA viruses which have traditionally been thought of as a single species of the *Orthopoxvirus* genus [14]. This monophyletic convention has endured in the most current International Committee on Taxonomy of Viruses (ICTV) official report. ICTV nomenclature changes for CPXV are awaiting proposals and pending “results of a wide-scale genome sequence study” [15] despite research supporting CPXV species distinctions since 2004 [16].

A lack of viral distinction due to the absence of concepts such as viruses and genomics has historically plagued the designation “cowpox virus”, once ascribed to any pox-forming disease derived from cows (*Bos taurus*). Attempts starting in the late 1930s to bring order to the designation [17] have inadvertently expanded the name to include poxviruses, which we now believe to be members of at least five unique clades, likely representative of different viral species. Unifying characteristics of these genetically distinct CPXVs are a Eurasian distribution, suspected rodent reservoirs, large genomes, and broad host ranges. This broad host range is of increasing concern as cases of fatal and nonfatal CPXV infections in non-endemic zoo animals, farm animals, and humans are on the rise [18,19,20], with ~40% of documented human cases occurring in a single modern outbreak between 2008–2011 [21]. While human cases are still relatively rare and only appear fatal in immunocompromised individuals, the need to better understand and define these viruses is apparent.

Today, CPXVs represent a zoo full of viral diversity in need of reclassification. Herein, we attempt to clarify the chronology of the nomenclature that follows the cowpox virus name to the present day. We provide a detailed review documenting the complete account of CPXV identity from 1768 to the 21st century, clarifying how the diagnostic criteria for CPXVs have evolved with scientific revolutions and CPXV-specific discoveries. Subsequently, we examine the phylogenetic data that argues for species-level distinctions of CPXV members, contextualizing the effort to rename CPXVs as unique species, discussing one possible naming proposal and the need of future nomenclature. We then focus on the current CPXV research, looking at the functional characterization of CPXV genes and focusing on host–pathogen interactions, including experimental infections and animal models of CPXV attempting to distinguish phenotypic differences of different CPXV species. Also covered is CPXV pathology in animals and humans, surveys of CPXV reservoir hosts, geographic range, modern outbreaks in various hosts, and finally the potential threat of CPXVs to provide a comprehensive review of modern and ancient CPXVs.

## 2. Variola Virus, Variolation, Cowpox Virus, and Vaccination

The complete historical account of CPXV begins with another virus: VARV, the causative agent of smallpox [22]. For millennia, humans have been afflicted by VARV with evolutionary analyses putting its origin around 2000–1000 BCE in the eastern part of Africa [22]. Although it has not been confirmed with molecular techniques due to a lack of preserved samples, it is assumed that VARV most likely caused many of humanity’s early plagues, with past contemporary texts highlighting overlapping symptoms with more recent VARV outbreaks, such as the Roman Empire’s “Great Pestilence” or Antonine Plague in 165 CE [23] and descriptions of smallpox-like symptoms in China 340 CE [24]. Increased trade and contact between civilizations led to smallpox-like outbreaks spreading throughout large parts of the world, with epidemics in early civilizations in Europe, Africa, and Asia [24,25]. Outbreaks devastated civilizations, such as the Japanese smallpox outbreak of 735–737 CE which killed a third of the island’s population [26]. Later in the 15th and 16th centuries, VARV outbreaks aided western colonizers in their invasion of North and South America, as the disease they brought from Europe had never been encountered by the native people who were immunologically naïve, causing over 3 million native people’s deaths [27]. Due to its long-term association with humanity, worldwide spread throughout the ages, and the high mortality rate of VARV major strains of between 10 and 30% [28], many estimates have placed smallpox as one of the deadliest human pathogens ever, with some claiming a death toll higher than all other infectious disease combined [29].

After centuries of global death, VARV outbreaks led to the discovery of basic immunological principles such as immunological memory, with the inciting observation that smallpox survivors were protected from subsequent infection [30]. This observation combined with attempts to transmit less virulent versions of VARV led to the invention of variolation or inoculation, with records claiming variolation techniques back to around 1000 CE in China [24,31]. The knowledge was later exported as it was first documented outside of China around 1549 and around 1581 in India, though variolation in India might have been practiced much earlier [24,32]. Variolation in China was carried out by blowing smallpox scabs, after ritualistic treatment of scabs with various herbs and salts, into the noses of patients [33], while in India the technique consisted of using a “fine sharp-pointed thorn” to pierce an infected patient’s smallpox pustule then using the thorn to gently puncture the recipient’s skin [34]. These techniques resulted in a less severe smallpox infection and generally robust protection against future infection but had risks including death and the potential to cause outbreaks, which was a concern of future adopters of these techniques [35]. Yet, the procedure spread across Asia and westward to the borders of the Ottoman Empire and into Africa. The Indian forms of variolation were advocated for in the west by Cotton Mather and Zabdiel Boylston in the 13 original British colonies in America and by Lady Mary Wortley Montague in Britain [36]. In Boston during the early 1720s, Mather, who was informed about smallpox variolation by literature, and his slave Onesimus, who was inoculated in Africa, tried to crusade for variolation but it failed to gain widespread acceptance in the early puritanical America of the 1720s. Mather’s proposed inoculation from infected smallpox patients to healthy individuals was attempted by Zabdiel Boylston, who was initially skeptical but possibly convinced by the intensity of the 1721 smallpox epidemic in Boston. The procedure was successfully conducted on his own son and the children of his slaves. Boylston was then elected to the Royal Society of London in 1726 after the procedure gained some acceptance in the colonies [37]. Around the same time, Lady Mary Wortley Montague, a wife to a British diplomat stationed in the Ottoman Empire and mother of two young children, learned of the technique through a different source, a local Greek practitioner [32]. Caring for her children’s safety, she had them inoculated against the disease in 1721 as at the time VARV was endemic to Europe where it overwhelmingly infected and killed children. Her advocacy after their return to the British Isles and the subsequent dissemination of the technique with Boylston’s help led to its wide adoption amongst physicians and made it a great success, with a 10x lower case fatality rate of variolation versus natural infection [38].

Throughout Europe, an increasing demand for inoculation dispensaries caused the procedure to be well known in the West [39]. It is during this period that the first mention of recorded cow-pox emerged. Pre-1768, cow-pox was recorded as a relatively rare benign disease acquired by milkmaids while milking infected cows in western England and Europe [40]. This origin has been contested in the modern day with arguments that claim the first accurate record of cow-pox came from John Fewster, who was first informed of cow-pox by farmers in 1768 (discussed further below) [41].

The true origins of the first cases of cow-pox are lost to time, but Jenner claimed that dairy workers in the area at the time were frequently exposed to cow-pox as it was “known among the dairies time immemorial, and that a vague opinion prevailed that it was a preventative of the small pox” [10]. Named after one of its hosts, cows, cow-pox infection generally resulted in a few typical vesicular pox lesions, a mild fever, and led to the surprising observation from afflicted milkmaids and/or farmers of long-term protection from smallpox infection. From observations of this phenomenon, a few individuals, including Benjamin Jesty and Peter Plett, began working on inoculating children, the most susceptible group to smallpox, with cow-pox [30,42]. The most famous among them was Edward Jenner, the physician credited with the scientific documentation and dissemination of vaccination; though, his path to this discovery was built on the work of others, which even he acknowledged [30,41]. One such contributor was the British physician John Fewster, who claimed in 1768 to have noticed a patient who had no response to VARV inoculation [43]. Upon questioning, the patient revealed that he had never had smallpox but had been infected with cow-pox. Fewster allegedly connected the two ideas and believed that cow-pox may be protective against smallpox infection, urging others to investigate for themselves at a medical society dinner that was attended by a young Edward Jenner [44]. Years later, Jenner experimentally tested this hypothesis by infecting a child, James Phipps, with a sample of cow-pox from a milkmaid named Sarah Nelmes, which resulted in a single pustule and mild fever. He then challenged the young boy six weeks later by infecting him with a sample of smallpox to which the boy had no reaction to [11]. These efforts led to the discovery of the cross-protective nature of orthopoxvirus infection, a rare viral/immunological phenomenon, wherein infection from one member of the genus generates immunological memory that protects from infection by other genus members [45]. From there, Jenner popularized vaccination with cow-pox as a safer alternative to variolation after a presentation to the British Royal Society, his self-publication of his experimental findings [11], and a subsequent world tour championing cow-pox vaccination [46]. These efforts paved the long road to the eradication of smallpox through surveillance, control initiatives, and worldwide vaccination campaigns with VACV in the 20th century, led by the World Health Organization (WHO) which on 8 May 1980 at the 33rd World Health Assembly declared natural VARV eradicated [3].

## 3. Post Jenner: Vaccinia and Cow-Pox Intertwined

In Jenner’s report, he attempted to give an alternative name to cow-pox in the title of his self-published work *Variolæ Vaccinæ* [11]. *Variol*æ is derived from the medieval Latin name of smallpox, *variola*, while *vaccinæ* was based on the Latin word for cow, *vacca*. Soon after, Jenner’s newly created process was translated into French as “vaccin” and “vaccine”, and subsequently the process of using a “vaccine” on a patient was named vaccination in English [47]. Vaccination is now used to describe all modern inoculations to illicit immunity thanks to Louis Pasteur’s attribution of “vaccination” to his own protective inoculations [48]. However, while Jenner’s naming of the *variolæ vaccinæ* did not catch on, the New Latin for cow-pox, *vaccinia*, did, creating over two centuries of confusion and mystery regarding the origin and naming of the modern “cowpox” virus, VACV, and a third potential source of the vaccine, horsepox virus, which is still under investigation today [49,50,51].

Horsepox virus further obscures the origin of the vaccine virus; Jenner originally believed his cow-pox strain, *vaccinia*, may have originated from horses in which it caused a now-extinct disease known as grease, and it spread between farm workers [11]. He continued to privately believe a potential equine origin, comparing the agent that caused grease in horses to the efficacy of his vaccinia vaccine and observing no major differences. Unknown at the time was both the concepts of viruses and the cross-protection afforded to other *Orthopoxvirus* genus members by vaccination with another orthopoxvirus, making it difficult to evaluate claims of distinct poxviruses of that era [52,53]. Of note, the classification of these viruses by host also makes claims of the modern relation of these ancient viruses dubious without samples for genomic testing, and this naming convention has haunted poxvirus designations to the present day. In 1817, Jenner provided a stock of equine origin pox vaccine to the national vaccine establishment further obscuring the origin of modern VACV and CPXV naming conventions [40].

Additional complicating facts include the propagation method of the multiple stocks of smallpox vaccine each obtained from various sources. For the first 80 years of smallpox vaccination, stocks were propagated arm to arm, usually using young children and orphans who were shipped across Europe and the Americas. Many contaminations were recorded from this method as issues with maintaining infection in a human supply chain led to unhygienic practices such as sourcing of infectious pox from patients afflicted with other maladies. This led to recorded contaminations of vaccines including syphilis and hepatitis, which were then unknowingly used to infect others [46,54].

Natural outbreaks of cow-pox continued to occur into the 20th century with cases throughout Europe and even claims in North America, although with decreasing frequency in light of widespread vaccination [55,56,57]. From these outbreaks, more vaccine innovations emerged. Italian and French efforts led to the propagation of vaccine stocks in cattle and horses, with one cow-pox source being the Beaugency lymph in 1866 cultivated from cattle in Beaugency, France, and horsepox from a horse from an 1880 outbreak in the Haute-Garrone department of France [57,58]. While greatly improving vaccine stock quantity and safety from human pathogens, these vaccine sources lead to other zoonotic contaminations, especially earlier in the 19th century. Viruses that caused lesions on cow udders, now thought to be pseudo-cowpox (milker’s nodules) and herpes mammillitis, may have caused contamination of vaccine stocks and vaccination with these other diseases as some affected stocks did not immunize humans against smallpox [54]. Scientific discourse at the time also blurred the definition of what we now know to be distinct viruses with distinct host ranges as many studies attempted to turn smallpox into cow-pox, usually by transferring smallpox to cows and then inoculating children, generating questions regarding attenuated smallpox or other orthopoxvirus contamination as an origin for vaccine stocks [59,60] though many of these transmissions to cows were unable to be further propagated and when successful resulted in smallpox in children [61,62]. While some of the smallpox vaccines’ origins and distribution are well documented, such as the Beaugency lymph [58], the virus or viruses behind these vaccines are obscured by time. In the confusion of the various naming conventions, the apparent chronology is that some of these “cow-pox” materials were used in vaccines and became ubiquitously known as vaccinia vaccine, and later VACV, with years of intentional passaging in humans, cows, and horses, and later cell culture cells affecting its genetic makeup, while natural outbreaks of pox disease in cows became known as cow-pox and later CPXV if not used in vaccines.

## 4. From Poison to Intercellular Obligate Parasite

Despite pox diseases being referred to as viruses (as in its Latin usage, “poisons”) even in Jenner’s day when referring to diseases of different etiologies [10], the discovery of and modern conceptualization of viruses as agents of infectious diseases in 1892 by Dmitri Ivanowsky (Ivanovsky) [63] and in 1898 by Martinus Beijerinck [64] revolutionized research and birthed the field of virology [65]. This drastically changed scientific thought and the core conception of CPXV identity as well as that of other poxviruses. This initial distinction by Ivanowsky separated viruses from bacteria based on their filterability in porcelain filters with extremely small holes, with viruses being much smaller than traditional bacteria which allowed viruses to pass through. The work of Ivanowsky and Beijerinck also demonstrated that viruses could not be propagated without a host, eventually leading to the definition of viruses as obligate cellular parasites. This change in dogma greatly affected research into poxviruses, shifting efforts to microbiological investigations of CPXV, VACV, and VARV, although work to define the microscopic nature of poxviruses had begun even before the conceptual birth of viruses with the first microscopic observations of vaccine lymph by Buist, who believed the “micro-organism” in his observations to be “spores” [66], later to be called elementary bodies [24,67]. Despite difficulties in elucidating animal virus structure [68], by 1956 the virion structure of VACV was identified by electron microscopy [69]. These developments greatly aided the structural understanding of poxviruses but shed little light on differentiating between poxviruses as they all possess the same general brick-like structure [19]. To begin discerning poxvirus species from one another would take more discriminating approaches and would begin the path to untangling cowpox naming conventions.

The first recorded approach to characterize the histological and molecular properties of VACV and CPXV was advanced by Alan Downie in 1938 [17]. Downie starts this historic paper by first highlighting the state of uncertainty regarding the terms of vaccinia and cowpox in his day: “most strains of vaccinia, whatever their origin, have become stabilised in their virulence and immunological characters and although the terms vaccinia and cowpox are frequently used synonymously it seems doubtful whether this practice is justified.” He also defined “Cowpox virus” as strains isolated as “which have been isolated from the spontaneous disease in cattle or from lesions in man caused by infection directly from that source,” highlighting the misleading host-based naming convention of poxviral diseases. Using the CPXV strain he identified in Downie et al. [70], the prototypic modern CPXV strain Brighton Red (BR), isolated from infected farmers in Brighton, England, Downie began the process of distinguishing poxviruses.

After looking at infected tissues under a microscope, Downie identified differences in inclusion body (protein aggregates) presentation with only CPXV infection presenting unique type A (acidophilic) or A-type inclusion (ATI) bodies in infected cells, later named Downie bodies by others [71]. Observations of infections of the chorioallantoic membrane (CAM) of chicken eggs further distinguished CPXV and VACV infections, as CPXV infections resulted in a “haemorrhagic character”: hemorrhagic lesions or red bloody pock marks. Although future experiments comparing many orthopoxviruses via similar methods [72] identified more diversity in CPXV lesions on CAM, such as opaque white lesions and variations of hemorrhagic character that caused a shift from red pocks to white pocks, they caused some difficulty in classification and would be investigated later on [73,74]. As such, the ATI bodies identified by Downie, seemingly unique to CPXV, were subsequently used to identify CPXV infection, although later research found the presence of ATIs in other poxviruses [75]. These initial distinguishing phenotypic characteristics and those found in other poxvirus members [76] helped to lift the veil of poxvirus identity especially between VACV and CPXVs, providing a method to distinguish poxvirus infections. One month later, Downie published a paper looking at immunological differences between VACV and CPXV, finding differences in heat-labile antigenic compositions though both infections provided immunity against the other in rabbits [77].

Downie’s CPXV characteristics, specifically ATI bodies, were used in conjunction with other methods, including ceiling temperatures for optimal viral growth on CAM, virus host range, serological relationships, antibody cross-protection experiments, plaque morphology in tissue culture, dermal reaction in rabbits, and polypeptide analysis to identify CPXVs and other orthopoxviruses until near the end of the 20th century [78,79,80]. However, the most popular method of viral species determination for CPXVs was the presence of ATI bodies. Starting in the 1960s, many new CPXVs were identified by some of these various methods as these infections went from rare infections in cattle to deadly infections in diverse mammalian zoo animals located throughout Europe [81,82].

Between 1960 and 1986, 21 cases of CPXV infections were identified using ATIs as a major discerning factor [83]. Infected okapi at the Rotterdam zoo in the Netherlands were diagnosed by the presence of type A inclusions with CPXV infections in 1971 [84]. In Russia in 1977, one of the first documented cases of poxvirus infections of *Felidae* family members including lions, cheetahs, ocelots, and many more species, including a human animal handler in a zoo, was linked to a potential CPXV through ATI bodies, plaque morphology, and neutralizing antibody tests [85], and was later considered a CPXV member [82]. In the same year, attempts were made to distinguish an elephant-derived “elephantpox” from VACV and CPXV, finding it had antigenic properties of both and different growth temperatures, placing “elephantpox” uniquely in its own camp [86]. Despite the author’s classification, this elephantpox was erroneously thought to be a “cowpoxlike” virus by others [87]. Elephantpox virus would continue to be referenced and connected to CPXV until 1999 in the research literature [88], where it then appears to be subsumed into the cowpox virus name and has since fallen out as a naming convention as there are no current poxvirus designations officially recognized as “elephantpox virus” by the International Committee on Taxonomy of Viruses (ICTV) [89].

As late as 1990, the sole presence of ATI bodies and antibody neutralization assays for orthopoxvirus-specific antibodies was used to identify CPXV in an infected human patient and their pet cat [90]. While these methods helped in the classification of poxviruses, they simultaneously and erroneously expanded the cowpox name to many poxviruses possessing common phenotypic traits. As time progressed, Downie’s established distinction between poxviruses, especially CPXVs, became more nebulous and blurred as more poxviruses were interrogated as to their species and presented with characteristics related to CPXV and many other poxviruses in diagnostic tests [19,91]. In 1971, the precursor to the ICTV established CPXV as one of ten virus species in the *Poxvirus* genus [92], later updated to *Orthopoxvirus* genus [93]. The limited amount of orthopoxvirus designations potentially contributed to attempts to subsume unique European orthopoxviruses into the cowpox name based on their type A inclusion bodies. Many of the above-described viruses were categorized as CPXVs in the book/review *The Orthopoxviruses*, possibly influencing their scientific conception and acceptance as one species [94]. However, during this era, the fundamental unit of poxvirus distinction, viral genomic DNA, and the technology behind genomics was progressing and would soon revolutionize the definition of what constitutes a poxvirus species, revealing the genetic diversity constrained under the cowpox name.

## 5. DNA and Genomics: Cowpox Virus to Cowpox Viruses

The next revolution in biology, the discovery of the role of deoxyribonucleic acid (DNA) and the subsequent genomics era, completely changed the core concept of viruses and allow for the distinction of viral species to be made on the basis of their simplest defining feature: their unique DNA/RNA sequence [95] (with some breakdown of the concept of viral species with the description of viral quasispecies [96]). The discovery of the structure of DNA in 1953 [97,98] and research demonstrating DNA as the biological basis of heritability and protein expression, especially in viruses, was well supported by the experiments of the members of the famous “Phage Group”, including the Avery–MacLeod–McCarty experiment [99]. While the complete historical evolution of the central dogma of biology DNA–RNA–Protein is well documented [100] with many research groups contributing, the focus on the initial genetics of poxviruses was from the start restricted by a lack of molecular tools. While DNA as a component of poxvirus virions was first identified in the 1940s [101], it took until 1967 to determine that poxviruses were identified to have dsDNA linear genomes [102,103,104]. In 1982, Baroudy et al. discovered the intricacies of the incompletely base-paired flip-flop terminal loops of VACV dsDNA establishing modern models of orthopoxvirus genomes [105]. However, early genetic analysis of viral genomes began with the discovery and application of restriction enzymes (RE), DNA endonucleases that cleave DNA at unique sequences first identified by observing differences in phage replication in certain strains of *Escherichia coli* [106,107,108,109]. These molecular scissors allowed for the comparison of viruses based on the various lengths and numbers of fragments of DNA resulting from restriction enzyme digests of viral DNA visible after loading the DNA fragments on agarose gels and subsequent electrophoresis [110].

CPXV BR was first genetically described in 1978 by restriction enzyme digest comparing 12 different isolates of orthopoxviruses including VACV, VARV, and monkeypox virus, and provided a genetic methodology to distinguish these viruses [111]. CPXV BR was shown to have a genome 23 to 29 megadaltons larger than VACV strain DIE [112]. In 1985, a paper updated the orthopoxvirus restriction profiles and maps, looking at 38 unique viral genomes identifying CPXV BR as having a large genome and a unique restriction fragment pattern [113]. Orthopoxviruses classified by others as CPXVs in Germany, isolated from elephants, rhinoceros, okapis, and other mammals reported in 1986, were genomically analyzed by restriction enzymes *BamH* I, *Mlu* I, *Nco* I, and *Sal* I resulting in robust viral strain differentiation [87]. This work demonstrated genetic differences between all tested isolates highlighting the value of RE digests to differentiate viruses. Unique for the time, the authors refrained from making attempts to classify the isolates, focusing on the genetic diversity their method revealed; though, in future papers they would support some of the viruses’ inclusion as cowpox or “cowpoxlike” viruses due to “close relatedness” [83]. Work by the same group in 1987 classified one human orthopoxvirus infection into a “cowpoxlike” group, finding more unique orthopoxvirus genetic diversity by RE digest [114]. Early attempts to more robustly compare these restriction enzyme profiles began in poxviruses without the inclusion of CPXV strains, using a method to convert RE fragments into binary data and then run through early computers to create dendrograms attempting to demonstrate relationships between viruses [115]. Later, only the BR strain of CPXV was compared to diverse orthopoxviruses using this method, resulting in a clade including ectromelia virus, CPXV, variola major virus, and variola minor (alastrim) virus [94].

While early genomic analyses including CPXV mainly focused on differentiating orthopoxviruses from one another, cowpox-centric approaches began to reveal the diversity of viruses constrained under the CPXV name. A comparison of orthopoxviruses recovered from cats and cattle in Britain in 1991 combined the methods of restriction enzyme digests and the identification of hemorrhagic lesions in CAM and type A inclusion bodies to identify all the unique isolated viruses as CPXV, with “minor differences” in these genomic samples’ RE fragments size [81]. In 1997, an unknown orthopoxvirus that caused a lethal infection in an immunosuppressed 18-year-old man was analyzed by pock morphology on CAM, dermal infections of rabbits, and RE digest of polymerase chain reaction (PCR) [116] fragments, thus placing the unknown virus in the CPXV ranks [117]. ATI and RE digests also identified four CPXV isolates from Norway and Sweden in 1998 that were added to the cowpox name, despite genetic differences to CPXV BR [118]. In 1999, a characterization of 14 orthopoxviruses from humans and animals in Germany identified them all as CPXV variants using CAM hemorrhagic lesions, anti-type A inclusion protein Western blots, and RE mapping, which still identified genomic differences including a unique 4.0 kb Hind III fragment found in five of the isolates [117]. These studies revealed genetic diversity in these CPXVs, unfortunately; instead of distinguishing new viral species or attempting to further characterize these genetic differences, the lack of discerning tools led to continual growth in the CPXV ranks due to the presence of ATIs and similar enough RE digest DNA profiles.

Attempts to further improve viral genetic distinctions came early in the genomics era with the efforts of Sanger, whose group’s research led to the invention of Sanger sequencing of DNA throughout the late 1970s [119,120]. This innovation liberated biologists from phenotypic classification and other methods to discern and classify organisms and viruses into species and begin to more rigorously categorize species based on the sequence of their genes and genomes. Sequencing was combined with PCR of RE fragments and whole genes in addition to other molecular cloning techniques to revolutionize our understanding of genetic diversity and start to organize the relationship between poxviruses once again.

The first use of these techniques with CPXVs was in 1982, where the sequencing of a *Sal* I fragment (2725 base pairs) of the inverted terminal repeat (ITR) regions of the CPXV BR strain cloned into plasmids, revealed three unique regions flanking two sets of repeating sequence sections made of four subunits [121]. Comparison to the VACV terminal regions demonstrated that the VACV repeat sections only possessed one subunit in common with CPXV. Further sequencing of RE fragment plasmids of CPXV BR and its white pock variants by the same group revealed that 9/10 white pock variants had the right-hand terminal region ITRs replaced with a 21–50 kilobase (kb) pair left-hand terminal from the same genome, while the 10th variant had a deletion of 12–32 kb in the right-hand terminal region [74]. Later the group identified the gene, missing in the white pock variants, responsible for red pock marks in CAM as related to plasma proteins that are inhibitors of serine proteases involved in blood coagulation pathways [122]. Throughout 2000–2008, 72 samples of CPXVs from cats in Germany were PCR amplified for their hemagglutinin (HA) genes and analyzed to make a phylogenetic tree supporting four different CPXV groups with many subclades for a single gene indicative of great genetic diversity [123]. During 2009, an outbreak of CPXV from infected pet rats (*Rattus norvegicus*) to humans in Compiègne, France, provided another opportunity to determine CPXV isolate HA differences [124]. A smaller phylogeny with the HA gene sequences of various orthopoxviruses and the new HA sequences from the CPXV outbreak supported large distances between CPXV GRI-90, which formed clades close to VACV, while CPXV BR and the novel CPXV strains formed clades outside the other orthopoxvirus HA sequences with ectromelia poxvirus in seven unique subclades.

Efforts to further investigate genetic diversity required sequencing and analyzing longer fragments of orthopoxvirus genomes. As tools improved, CPXV genomes began to be analyzed in this regard. In 1998, the left 52,283 bp and right 49,649 bp terminal regions of CPXV GRI-90 were sequenced, identifying 88 ORFs [125]. Amino acid comparisons between six genes from both CPXV BR and CPXV GRI-90 revealed a range of 82–96% sequence identity. Further analysis of CPXV sequences and other orthopoxviruses uncovered that the sequenced CPXV-GRI possessed the most intact repertoire of immunomodulatory and host range genes, leading the authors to hypothesize that CPXVs may be similar to an ancient last common ancestor of orthopoxviruses, although future research demonstrated huge genetic disparities in CPXVs, including analyses of North American orthopoxviruses [126,127], making claims of unified orthopoxvirus ancestry to extant CPXVs unlikely as no single extant CPXV seems ancestral to modern orthopoxviruses based on evolutionary analyses. Rather, it seems that the unifying genetic features of CPXVs are large genomes and gene sets that have been lost during host adaptation in other viruses such as VARV and VACV. One possible explanation in need of experimental support is that modern CPXV clades possess more “complete” repertoires of genes related to maintaining their broad host ranges compared to other orthopoxviruses. As a result, they appear more like the orthopoxviruses’ last common ancestor compared to other orthopoxviruses with more restricted host ranges such as VARV and lab-passaged VACV, which may have undergone “reductive evolution” through gene loss [128,129,130].

A major advance for poxvirus virology was the full sequencing of the VACV genome by Goebel et al. [131] in 1990, which at the time was an expensive and labor-intensive effort achieved by sequencing VACV RE fragments inserted into plasmids. By 2003, many poxviruses had been fully sequenced allowing for genome-wide comparison of gene sequences. Twenty-one poxvirus genomes were uploaded online, including CPXV BR which at the time had not yet been published, and were analyzed as to their gene content and evolutionary relations to one another [132]. This effort was made possible by the creation of the Poxvirus Bioinformatics Resource (now https://4virology.net/ (accessed on 6 of February 2023)), one of the first viral bioinformatics websites aiming to facilitate research into poxvirus genetics and other tools such as the Poxvirus Orthologous Clusters (POCs) tool allowing for easy analysis of genes, promoters, and orthologs from poxviruses [133]. Though CPXV BR was not deeply analyzed in the paper, its full genome and the new genetic bioinformatic tools available would contribute to further studies.

An evolutionary relationship-based approach was built on the uploaded sequences in 2004 [16]. Twelve full genome orthopoxviruses were analyzed to generate phylogenetic trees using 12 terminal region genes shared by all the members. This approach accounted for the variable relationships observed when generating phylogenies from single genes previously observed. CPXV BR and CPXV GRI-90 were found to cluster differently, with CPXV-GRI grouping with VACV Copenhagen, modified vaccinia virus Ankara (MVA), monkeypox Zaire, and ectromelia virus strains, while the CPXV BR clade was located between this group and another branch with VARV and camelpox virus, strongly indicating the two are separate viral species in need of reclassification as a unique species.

In 2011, Chasing Jenner’s Vaccine: Revisiting Cowpox Virus Classification [134] was published, taking in years of CPXV research, modern genomic analysis, and increasing observations of CPXV diversity to attempt to change the paradigm regarding CPXV classification conventions. Genomic sequences from 12 isolates previously classified as CPXVs including CPXV BR, GRI-90, and isolates from France, Germany, Finland, Norway, the UK, and Austria were analyzed phylogenetically. These genomic sequences were reduced to include the complete coding region between C23L-B29R and then aligned and used to construct a phylogenetic tree. Part of this analysis included a threshold value based on taterapox virus and VARV patristic and genetic distances which were used to distinguish between viral species within the paper as the two viruses are “currently recognized and undeniably distinct OPV species.” Using this threshold, five unique clades of CPXVs were identified with species-level genomic distinctions. Additional genetic analysis supported this species distinction by the alignment of three specific CPXV genes: CPXV_BR_021, an epidermal factor-like protein; CPXV_BR_191, a tumor necrosis factor receptor homolog responsible for antagonizing tumor necrosis factor alpha (TNF-α); and CPXV_BR_212, an interferon (IFN) α/β receptor homolog responsible for IFN α/β antagonism; and by identifying unique clade-specific non-synonymous mutations for three of the clades (1,3, and 5) in some of the aligned genes. This provided a stronger framework to distinguish CPXVs from each other than ever before and began the advocacy for a CPXV reclassification in earnest. The authors maintained, however, that if these relationships are correct then it is likely that there are biological similarities beyond genetics that unite the clades/species, such as pathogenesis, including phenotypic distinctions during infection and host range that should be investigated to generate robust species designations. However, their results substantiated the diversity of viruses under the cowpox virus name, observed by other researchers but never explored in such depth and focus with regard to species delineation, providing a taxonomic context that revolutionized thinking about the diversity of CPXVs.

Then in 2013, 22 novel whole genomes of CPXV isolates from clinical cases involving humans and other hosts were sequenced by massive parallel pyrosequencing and then analyzed phylogenetically supporting the conclusions of Carroll et al. [134] that the CPXV name does not represent a monophyletic group [135]. In their methodology, the authors identified conserved genes at different taxonomic levels, with 49 genes at the *Poxviridae* family level (PVC), an additional 41 at the *Chordopoxvirinae* subfamily (CVC) level, and an additional 48 at the *Orthopoxvirus* genus level (OVC) (all with ≥80% nucleotide sequence identity in all orthopoxvirus genomes analyzed) that were used in different combinations to generate phylogenies. The phylogenies of the PVC/CVC/OVC and PVC gene sets again supported five unique CPXV groups, though the two sets differed in their ordering of the three groups excluding the VARV-like and VACV-like CPXVs. They also examined the correlation between the conserved gene set-based phylogenies versus the historically popular HA gene-based phylogenies, determining that HA analyses failed to recapitulate the more robust relationships displayed by gene sets as single gene phylogenies are prone to generating various distinct evolutionary relationships.

Published in 2015, a study focused on the analysis of two new CPXV isolates: one isolated from a common vole (*Microtus arvalis*), CPXV FM2292, and CPXV RatPox09, a strain identified in a French 2009 pet rat outbreak [136]. These isolates were analyzed following the gene set of the VACV-Copenhagen strain including genes C23L-B29R [134], following the general pattern of clades established by Carroll et al. [134] and Dabrowski et al. [135] with five major groups of CPXVs, this time named: VARV-like, CPXV-like 1, CPXV-like 2, CPXV-like 3, and VACV-like.

In 2017, a thorough phylogeny of 83 whole genome orthopoxviruses including 20 novel CPXV isolates maintained that CPXVs are the only known polyphyletic *Orthopoxvirus* genus member [14]. Using a 142,286 bp region shared by all strains tested, this more up-to-date phylogeny identified four clades of CPXVs: VARV-like, CPXV-like 1, CPXV-like 2 (a combination of clades CPXV-like 2 and 3 [136]), and VACV-like, as well as three unique outlier CPXV isolates with one Ger 2010 MKY, called CPXV-like 3. A supporting consensus network using the sequences and combining 29 phylogenetic trees affirmed the clades while revealing potential signs of viral recombination. To further investigate potential recombination, genomic sequences were compared to one another through a bootscan analysis [137], picking out sequence regions with high levels of nucleotide similarity with other clades indicative of recombination.

Another study in 2017 [126] aimed to “consider the validity of the species *Cowpox virus* as currently understood given the new ICTV definition of virus species” in response to a 2013 paper announcing the ICTV’s new definition of a virus species: ”A species is the lowest taxonomic level in the hierarchy approved by the ICTV. A species is a monophyletic group of viruses whose properties can be distinguished from those of other species by multiple criteria” [138]. This 2017 paper interrogated CPXV as a species designation by briefly highlighting the history of CPXV distinction, examining the impacted nature of the CPXV name and by updating the current phylogenies [126]. With 9 supplemental whole genome CPXV sequences for a total of 46 CPXVs and other orthopoxvirus genomes, they created phylogenies that identified at least five clades (the same as identified by the other groups) but based on Bayesian posterior probability analysis and patristic and genetic distances, they proposed the possibility of the existence of 14 monophyletic lineages of CPXVs questioning the validity of the CPXV species as it still stands today.

Two recent investigations added knowledge to the CPXV diversity. The first study identified a unique strain named CPXV-No-H2, a mosaic virus that had undergone seven potential recombination events with several orthopoxviruses. The core genes formed a group with the previously identified CPXVs that are most closely related to the ectromelia virus [139]. The second study presented five additional CPXV genomes from CPXVs, which were isolated from cats and humans in Fennoscandia (the region including Finland, Norway, Sweden, and some parts of Russia), and subsequently determined their place in the *Orthopoxvirus* genus phylogeny [140]. Their analysis included 87 orthopoxviruses, once again supporting the five major clades of CPXVs consistently observed across studies. Although, based on genetic and patristic distance thresholds derived from the distance between TATV and CMLV (unlike Carroll et al. [134] which used TATV and VARV), they identified 18 “subspecies” of CPXVs surpassing their threshold, once again highlighting the species level genetic variation constrained under the CPXV name.

At the time of writing this review paper, there were 96 full genomes of CPXVs available at the National Institute of Health (NIH) United States National Center for Biotechnology Information (NCBI) database under the CPXV name and classification claimed in publications. They range from 196,570 bp to 229,131 bp (CPXV_K779 and CPXV/Rat Koelle, respectively) and come from diverse hosts, with the two largest groups being humans (29 isolates) and cats (24 isolates). This dataset is summarized in Table 1, indicating the CPXV phylogenetic classification, location, year of isolation, host, accession number, and references that describe the isolation or inclusion of the virus in the discussed phylogenies. Continuing efforts to completely sequence and characterize wild CPXVs will shed light on the evolutionary relationships between these viruses and aid in attempts to untangle their naming conventions.

Looking back on the knowledge we have gained through decades of research, it is clear that throughout history the cowpox virus name has encompassed a nebulous horde of viruses (see Figure 1). Since the first mention of cow-pox as a disease agent of cows, horses, and humans used in Jenner’s original vaccine, distinctions from other infections and viruses have been uncertain and difficult. Substantial efforts to distinguish CPXVs from other viruses led to Downie’s phenotypic distinctions and genomic analysis including restriction enzyme fragment analysis, modern sequencing, and phylogenetic trees, which have all better defined and discerned CPXVs from other orthopoxviruses and from the strains within CPXV taxonomic designation, revealing an abundance of viral diversity. During this era, scientists established poxvirus designations and expanded our collective understanding of the nature of these viruses despite limited tools and techniques, therefore advancing the field. Recent advances in sequencing technologies and increased availability combined with phylogenetic analyses have revealed the hidden diversity in CPXVs and have supported CPXVs as four or five major distinct clades of viruses. While these numbers of clades appear consistent over the years, these clades have been subsumed, merged, or split in the various analyses presented here. For example, the original five clades of Carroll et al. [134] are missing isolates that make up the VARV-like major clade of Franke et al. [14]. This additional factor makes it difficult to unify the various efforts to organize CPXVs. Patristic and genetic distance thresholds based on VARV and TATV and/or CMLV and TATV provide genetic evidence for additional species-level classification within some major clades supporting 14 [126] to 18 [140] possible subspecies as defined by genetic distances. However, these analyses did not take gene content, host range, or other phenotypic differences into account, which might aid the distinction of individual species. In attempts to rectify CPXV species distinction and naming, Mauldin et al. [126] proposed a nomenclature defining the five species-level clades of cowpox virus as “*Cowpox virus alpha*, *Cowpox virus beta*, *Cowpox virus gamma*, *Cowpox virus delta*, and *Cowpox virus epsilon*” to avoid unnecessary name changes [126].

A recent comprehensive phylogeny containing 82 CPXV genomic sequences and an additional 153 orthopoxvirus genomes, used a region of ~110 kb between OPG048 (also known as F4L or CPXV051 in CPXV BR) and OPG160 (also known as A32L or CPXV167) to identify five major CPXV clades [161]. Using this dataset, we performed phylogenetic analyses to include sequences from all 96 currently available fully sequenced CPXV genomes. This analysis showed a comparable topology to the one by Senkevich et al. and identified five major clades, which were well separated from the other orthopoxvirus clades (Figure 2, Appendix A).

The distinction of viral species is an area of great debate, with species demarcation being traditionally informed by discernible phenotypes. Movements to specifically define viral species based on evolutionary taxonomy, especially monophyletic clades, with distinction by multiple additional criteria such as host range are strongly advocated for and reviewed in [162] and the ICTV virus species definition [138]. The lack of clear distinctions between these viruses beyond genetics, including host range and other characteristics, is likely an important factor explaining why the ICTV has not reclassified CPXVs yet. While additional research will have to identify the unique properties that may define isolates belonging to unique species such as unifying genetics or gene content, pathogenesis, and host range, it is clear from these studies and decades of attempts to characterize CPXVs that there is a need for CPXV reclassification and new nomenclature.

**Figure 1 biomolecules-13-00325-f001:**
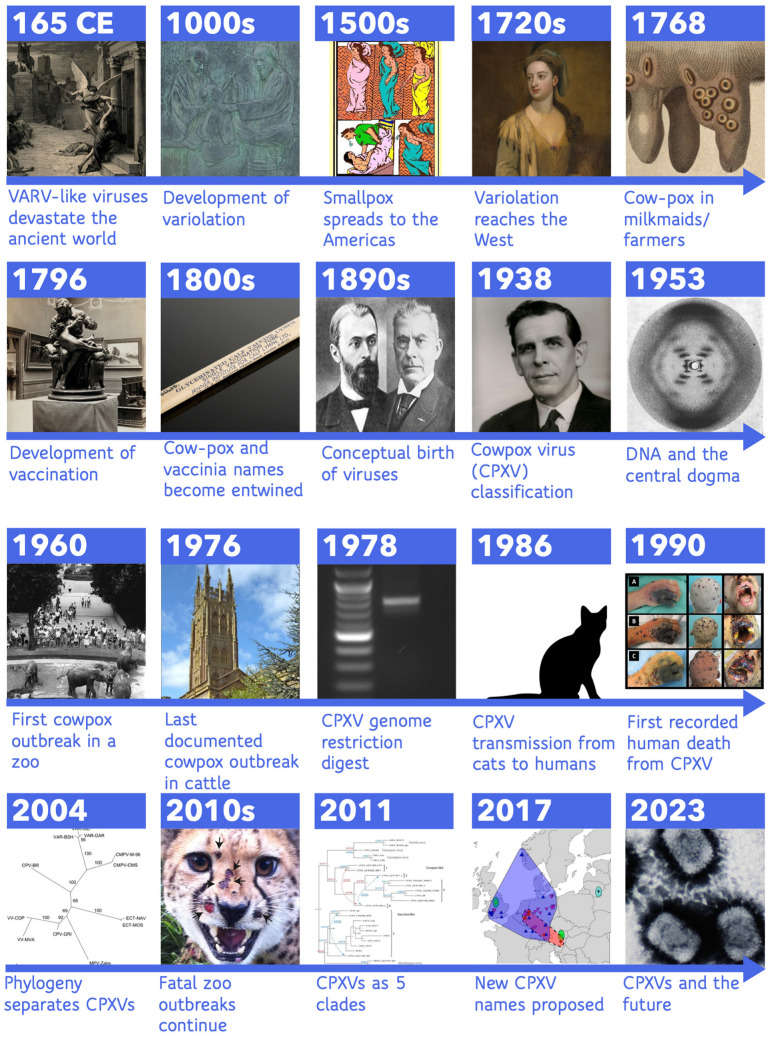
Timeline of major events in cowpox virus history. For more information see the indicated citations for each entry. 165 CE [23,24,25], 1000s [24,31],1500s [27], 1720s [32,36], 1768 [10,41,43],1796 [10,11,12], 1800s [49,50,51,52,54,58], 1890s [63,64], 1938 [17,70], 1953 [100], 1960 [82], 1976 [163], 1978 [111], 1986 [164], 1990 [154,165], 2004 [16], 2010s [151,158,160], 2011 [134], 2017 [126]. For image rights and licenses, see Appendix A.

**Figure 2 biomolecules-13-00325-f002:**
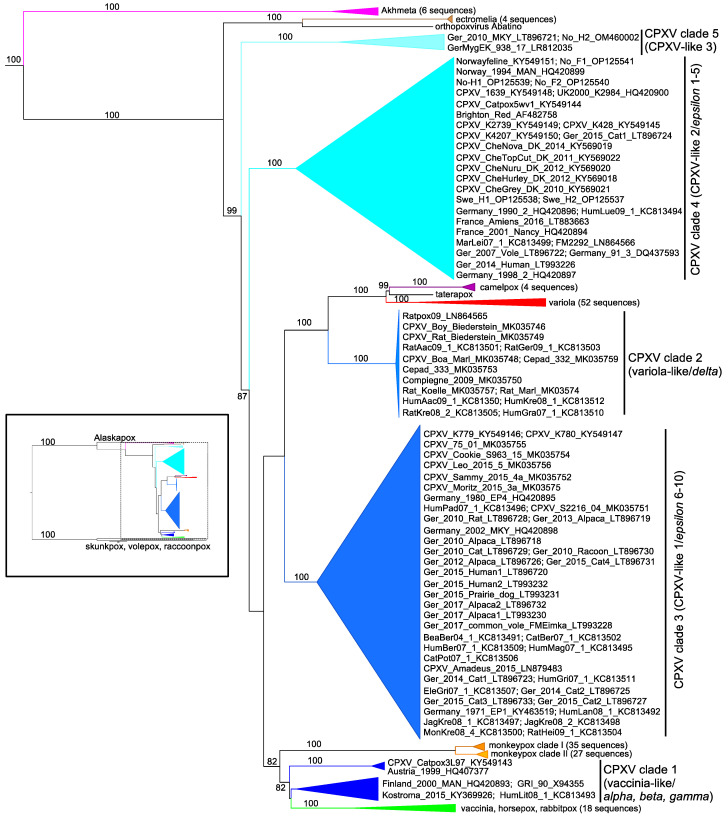
Phylogenetic analysis of orthopoxviruses. Multiple sequence alignment of genomic fragments ranging from orthopoxvirus genes (OPGs) OPG55 to OPG150, as described in Senkevich et al. [162] (see page 18 for an explanation of nomenclature), from 249 orthopoxviruses was generated with MAFFT v7.505 using default parameters [166]. A neighbor-joining phylogenetic analysis was performed using the Jukes–Cantor model [167] with nodal support assessed via bootstrapping (1000 pseudoreplicates) [168] excluding gaps and non-aligned regions. The resulting tree was visualized in FigTree v1.4.4 (http://tree.bio.ed.ac.uk/software/figtree/ (URL accessed on 6 February 2023)) and rooted to the skunkpox, volepox, and raccoonpox branch. For better clarity, the main figure concentrates on branches that form a sister clade to Alaskapox virus, while the inset shows the whole tree. For easier readability, the branches containing multiple-non-CPXVs from the same species were collapsed, and the number of virus isolates are indicated. The branches for the five major CPXV clades are shown as cartoons with virus isolates and accession numbers indicated. Bootstrap values >75 are indicated above/below the branches. Major CPXV clades defined by Senkevich et al. [161] can be found on the right hand side; in parenthesis are overlapping clade designations from Franke et al. [14] and Mauldin et al. [126].

## 6. CPXV Genes and Host Interactions

Despite the genetic diversity observed in CPXVs, one unifying feature is their large genomes with the highest number of genes in the *Orthopoxvirus* genus. For example, while poxvirus genomes are large for viruses in general, encoding over 150 genes, CPXVs possess ~210 genes on average [14,158,161]. Senkevich et al. [161] proposed the reclassification of homologous genes in the *Orthopoxvirus* genus as “orthopoxvirus genes” (OPGs) 1-214. Efforts have been made to unify gene nomenclature across the genus to simplify comparison of orthologs. This new system is used here where appropriate to designate genes from CPXVs. The large genomes and gene sets of CPXVs were once considered a unifying characteristic of designation, overturned by sequence analysis demonstrating genetic variability. Early estimates in 2006 placed the common set of poxvirus genes at 49 genes conserved in all sequenced poxviruses and 41 more present in all chordopoxviruses for a total of 90 conserved genes which code for proteins involved in key viral processes, such as replication, transcription, and virion assembly [169]. More modern analyses put the conserved gene count of orthopoxviruses at 138, with an additional 48 orthopoxvirus conserved genes identified (>80% sequence identify) compared to the chordopoxvirus gene set [135]. This well conserved gene set leaves around 50 genes that are either unique to CPXVs or have low sequence identity with other orthopoxvirus species genes. In orthopoxviruses, about half of the approximate total 200 gene products are involved in basic aspects of virus replication, while the remaining genes, which are also known as accessory genes, are involved in virus–host interactions, including modulation of the host immune response [161]. While VARV, VACV, and monkeypox virus encode half of the full set of accessory genes (53–60 genes), some CPXVs possess an almost complete set of accessory genes, for a total greater than 100 genes [161]. For an in-depth review of CPXV-specific immunomodulatory genes and their function please refer to Alzhanova and Früh [170].

One puzzling phenomenon observed in the *Orthopoxvirus* genus before the genomics age was the observed differences in host range despite viral similarities. For example, natural cases of smallpox (VARV) were restricted to humans, whereas VACVs and CPXVs demonstrated large host ranges infecting many species throughout the class Mammalia. While many factors are at play in determining the ability of a virus to productively infect a host species [29], one area of research has been at the interface of viral host range genes and host immune restrictors of viral replication [171]. CPXVs possess the largest set of intact host range genes in the *Orthopoxvirus* genus, which is thought to be related to their large observed host ranges in nature. Host range genes are genes that have been identified as important for successful virus replication in a subset of cells or host species, while being dispensable in others [171]. In CPXV BR, there are 26 identified host range family genes and 27 in CPXV GRI, with the extra CPXV GRI gene being CrmE, a TNFR II family inactivated by premature stop codons and deletions in CPXV BR [172]. These genes include orthologs of VACV K3L and E3L, Serpin family genes, p28-like, C7L family genes, T4, B5R-related genes, M11L/F1L, multiple ANK/F-box homologs, K1L, and TNFR II family genes covered in depth in [172]. The role of these genes in host range determination in vitro and in vivo is only just beginning to be understood. Here we present some additional studies on host–CPXV molecular interactions and experimental studies of CPXV virulence in various animal models.

Early molecular biology approaches focused on CPXVs began with forward genetic techniques to identify the gene and protein product responsible for the ATI body. A single protein, named 160K after its molecular mass, was identified as a key component of CPXV BR ATIs [173]. In 1988, another group confirmed the role of 160K in ATI formation [174]. Using early molecular biology tools and a CPXV strain CPR06 maintained in Japan [175], the authors isolated ATI-associated proteins and inoculated rabbits to generate antiserum that was used to detect 160K protein in Western blots. Then, using RE digests and gene linking, they identified an open reading frame (ORF) with a sequence matching the size of the ATI protein. The gene for the ATI protein was later named *ati* [176] (OPG 152/CPXV158) [161] and investigated for its role in distinct ATI–virion association phenotypes defined in [177]. Further work showed that the deletion of ATIs by the homologous recombination of OPG152 with green fluorescent protein enhanced CPXV replication in the lungs of mice, supporting the hypothesis that the lack of ATIs may be beneficial to poxviruses that rely on respiratory spread such as VARV [178], though more studies are necessary to define the role of ATIs in host pathogenesis and transmission.

In 1988, scientists noticed the differences in the host range of CPXV BR, which could replicate efficiently in Chinese hamster ovary (CHO) cells and VACV Copenhagen, which had earlier been deemed to have defective replication in CHO cells [179]. Genetic linkage experiments from CPXV/VACV recombinants revealed a host range gene that when sequenced coded for a protein of 77kDa for replication in CHO cells [180]. This gene was named CP77 (OPG23) [161] and subsequently was found to block TNF-α induced nuclear factor kappa-light-chain-enhancer of activated B cells (NF-κB) activation by binding to the p65 subunit of NF-κB and binding directly to Skp1 and Cullin-1 of the Skp, Cullin, F-box containing complex (SCF). Activated NF-κB translocates to the nucleus and activates pro-inflammatory transcription of immune genes such as IL-1, IL-2, IL-6, IL-8, IL-12, IL-18, CXCL1, CXCL10, TNF-α, and many others to recruit and activate leukocytes to sites of inflammation [181], while the SCF complex catalyzes the ubiquitination reaction of proteins, leading to their degradation in proteasomes [182]. CP77 was found to encode six ankyrin repeats at its N terminus necessary for binding to P65, and an F-box homology region (PRANC domain) necessary for SCF binding. The study demonstrated that while CP77 had to bind both p65 and SCF to inhibit NF-κB, deletion of the F-box region was not necessary for CP77-based host range expansion of VACV to CHO cells as both a VACV-possessing WT CP77 and another with CP77ΔF (F-box deletion mutant) were able to replicate efficiently, while VACV WT could not. Ten years later, CP77 was be found to be an inhibitor of sterile alpha motif domain-containing (SAMD) 9 and SAMD9L [183]. SAMD9 and SAMD9L are interferon-inducible cytosolic proteins with evidence for both antiviral and antitumor functions. Recent work has shown that SAMD9/9L binds to double-stranded nucleic acid for antiviral and antiproliferative function and may do so by inhibiting cellular protein expression when SAMD9/9L reaches high enough levels in the cell, though a mechanism is unclear [184]. Additionally, SAMD9 has been shown to interact with the cGAS/STING pathway to induce proinflammatory responses by type I IFNs [185]. CP77 was found to bind and inhibit Chinese hamster SAMD9L (ChSAMD9L) and rescue VACV replication in CHO cells, while CRISPR-Cas9 knockout of ChSAMD9L in CHO cells resulted in robust VACV replication [183]. Ectopic expression of ChSAMD9L in a human cell line, normally permissive to VACV, restricted viral replication. Taken together, these results indicate CP77 interactions with ChSAMD9L as responsible for the host range differences in CHO cells between CPXV BR and VACV Copenhagen; however, whether this interaction is important for host range in vivo remains to be seen. Comparisons of different CPXV strains’ CP77 orthologs showed a greater than 91% amino acid sequence identity, while all VACV strains have large deletions in their CP77 orthologs giving further insight into the differential host range.

Despite *Orthopoxvirus*-related genera (*Capripoxvirus*, *Leporipoxvirus*, *Suipoxvirus*, and *Yatapoxvirus*) possessing homologs of myxoma virus m153 protein, a major histocompatibility complex I (MHC I) antagonist that downregulates MHC I in a non-species specific manner [186], sequenced orthopoxviruses lack identifiable homologs leading to questions regarding these viruses’ ability to combat cytotoxic lymphocytes such as CD8+ T-cells [187]. In 2007, Dasgupta et al. [188] identified a unique CPXV BR MHC I antagonist that prevented MHC I release from the endoplasmic reticulum and was identified after observing differences between cell surface MHC I expression in VACV Western reserve and CPXV BR-infected cells. The same group later identified the responsible gene as CPXV203 (OPG195), which when knocked out surprisingly did not restore MHC I transport to the cell surface of infected cells [189]. Intrigued by this observation, the authors then identified an additional MHC I antagonist gene CPXV012 (OPG10a), which interferes with MHC I peptide loading by inhibiting the transporter associated with antigen processing (TAP). They showed that CPXV012 antagonized adenosine triphosphate (ATP) binding to TAP via its ER-luminal domain [190].

In 2020, CPXV virulence factors were identified from CPXV Ratpox09 [191]. After observing high lethality in infections of rats with this isolate, the authors discovered four Ratpox09 genes that are absent in CPXV BR. These genes were tested for their contribution to virulence as measured by mortality rates in Wistar rats. Insertions of the four CPXV RatPox09 genes into CPXV BR together increased the virulence as measured by lethality to beyond RatPox09 levels in Wistar rats. Single insertions of these genes independently into CPXV BR revealed one of four genes, 7tGP (CPXV030), resulted in a significant increase in virulence (close to Ratpox09 levels) by itself. Puzzlingly, this increase in virulence from a single gene was not linked with an increase in viral shedding seen in the Ratpox09 infection. While the function of 7tGP is yet unknown, the authors believe it to be an immunomodulatory protein that may affect T-cell function in hosts. The identification of this virulence gene is worthy of further investigation into the mechanism of this phenomenon.

A recent publication [192] from our lab has investigated one CPXV ortholog of VACV K3L, a gene responsible for inhibiting the antiviral host immune pattern recognition receptor protein kinase R (PKR) [193]. The CPXV Ratpox09 strain K3L ortholog in question, CPXV 043 (OPG41) [161], was found to be a potent inhibitor of multiple different mammalian PKR orthologs with only 2/17 of the species PKR tested demonstrating a resistant phenotype in a luciferase-based assay of PKR activation. The diversity of CPXV host range genes and their functional differences in vitro and in vivo is an area of research that must be further expanded to truly understand host–virus dynamics especially in light of the phylogenetic diversity of CPXVs.

## 7. Animal Models of CPXV Infection

CPXVs have been used in diverse animal models from rodents to non-human primates to study various aspects of pathogenesis. Often, CPXV BR was used in these studies as it is highly virulent in different laboratory strains of mice (*Mus musculus*) and used to test the effects of antiviral drugs [194]. Experimental CPXV infections in brown rats (*Rattus norvegicus*) to test the potential of the species as a reservoir host were conducted in the 1970s and 1980s [195,196], with later experiments designed to observe the pathogenesis of CPXVs derived from the 2008–2011 CPXV outbreak in pet rats (discussed in the outbreak section below) [145,197]. Other rodents including different vole species have also been experimentally infected to test hypotheses of reservoir host status and CPXV species viral characteristics [136,198,199]. Primate models of CPXV infection used common marmosets (*Callithrix jacchus*) [200] and cynomolgus macaques (*Macaca fascicularis*) [201,202] in attempts to model the typical and hemorrhagic presentations of human smallpox infection at biosafety laboratory (BSL)-2 containment levels. While all of these models have pushed our understanding of CPXVs and orthopoxviruses forward, host–CPXV interactions with different CPXV clade representatives are just starting to be examined in animal models. To highlight this area of research we present a few cases of animal model usage in determining CPXV species differences below.

To follow up on phylogenetic data that supports the existence of five CPXV clades [134], another group investigated five representative isolates of CPXV in vitro and in vivo to define their properties and the abilities of poxvirus antivirals to halt CPXV replication [203]. The representatives were CPXV BR (major clade 4), CPXV Germany_1980_EP4 (major clade 3), CPXV_FIN2000_MAN (called Finland_2000_MAN in Table 1, major clade 1), CPXV_AUS1999_867 (called Austria 1999 in Table 1, major clade 1), and CPXV_GER1991_3 (called Germany 91-3 in Table 1, major clade 4), and the mouse-adapted VACV Western Reserve (WR) strain. At the time, these five isolates were representative of the five clades described by Carroll et al. [134] but lack the genetic diversity captured by the designations later described by [14,161], missing the VARV-like CPXVs and the Ger 2010 MKY clade. Regardless, this marked an important step towards discerning CPXVs from one another. Initial viral replication assays in vitro in HEL cells did not reveal any major replication differences between CPXVs; however, in vivo infections did. The first distinction observed during infections of five-week-old female NMRI mice with CPXVs was differential host outcomes in mortality with three groups: 0% (CPXV_GER1991_3 (major clade 3) and CPXV_FIN2000_MAN (major clade 1)), 20% (CPXV_AUS1999_867 (major clade 1) and CPXV Germany_1980_EP4 (major clade 3)), and 100% mortality (CPXV BR (major clade 4) and VACV WR). Of note is the differential mortality across viruses within the same major clade, such as the large difference in mortality between CPXV_GER1991_3 and CPXV BR, which are both found in major clade 4. This observation may lend support to proposals to further divide CPXVs into more species, as there are likely distinct genetic differences within clades that result in differential viral properties. The ability for the viruses to successfully replicate in the lungs was correlated with lethality as well as high IL-6 levels in mice with an average of 1000 picograms/mL of IL-6 in the 100% lethal CPXV BR infections, which was much higher than in all other CPXVs tested. In support of this idea, increased host secretion of IL-6 in lethal CPXV infections had been reported before in CPXV-infected macaques [201,202]. In their discussion, the authors highlighted the potential for CPXV-mediated upregulation of IL-6 as beneficial to systematic spread as CPXV may use IL-6-attracted macrophages and monocytes to disseminate within the host [197,204], though severe tissue damage resulting from the cytopathic effects of CPXV infection could be the cause of increased IL-6 levels in more virulent infections as damaged and dying cells release IL-6 [205,206]. Additional cytokine profiling found no evidence of systemic TNF-α in any infection, most likely due to CPXVs’ large amount of NF-κB and cytokine antagonists [170]. The distinct properties of these viruses in mice were differences in lethality, virus replication in the lungs, and IL-6 levels, which are likely results of the genetic diversity of CPXVs and are worthy of additional studies in different animals with additional CPXV representatives.

In 2017, experimental infection of bank voles (*Myodes glareolus*), a potential reservoir host of CPXV, with diverse CPXV clade representatives was reported [198]. The isolates used were: CPXV BR (major clade 4), CPXV RatPox09 (major clade 2), CPXV Ger 91/3 (major clade 4), CPXV GER/2007/Vole (major clade 4), CPXV FM2292 (major clade 4), CPXV GER/2010/Cat (major clade 3), and CPXV FIN_MAN_2000 (major clade 1). Despite the genetic diversity of these CPXVs, all CPXV infections of bank voles displayed no clinical signs, and no viral shedding was detected in nasal or buccal swabs. Infection with all CPXV isolates, except for Ger/2007/vole, resulted in seroconversion after 21 days, and viral DNA was detected in the upper respiratory tract after infection with all strains but Ratpox09. These results supported the minimalist definition of reservoir hosts as infected vectors that transmit disease but show no clinical signs but failed to fit precise definitions of reservoir hosts that require pathogens to be permanently maintained in one or more connected host populations or environments before their transmission to other species [207]. A study supportive of these results in bank voles was recently reported, characterizing a bank vole-derived CPXV isolate CPXV GerMygEK 938/17 in experimental infections of banks voles, common voles, and Wistar rats [147]. This isolate was compared to CPXV GER2010 MKY from a fatal cotton-tailed tamarin CPXV outbreak reported in 2015 [122]. They found that GerMygEK 938/17 resulted in no clinical signs or mortality in bank voles, common voles, or Wistar rats and found positive tests of viral shedding only in the presumed reservoir bank voles. Ger 2010 MKY infections resulted in no clinical signs, fatalities, or viral shedding in common voles or Wistar rats, but bank voles all experienced 25% weight loss and 33% died as a result of their infections.

While animal models provide novel insights into CPXV pathogenesis and are starting to distinguish CPXV species diversity in some hosts, we still do not yet have clear phenotypic distinctions to create simple recognizable nomenclature of their host range or unique aspects of their pathogenesis. Further studies may struggle to define CPXVs in our currently limited pool of animal models, while dealing with viruses with such large host ranges and defining aspects of CPXVs may have to be aided by further experimental studies with these viruses.

## 8. Pathology of Modern CPXVs

Despite genetic diversity and a broad host range, the pathologic presentation of CPXV infection has shared characteristics albeit varied outcomes in its hosts. In humans and animals, the general disease presentation after viral infection is a single localized lesion that forms at the inoculation site where the virus was introduced through compromised barriers in the skin, micro-abrasions, or through bite wounds [208]. Viral replication in infected cells results in leukocyte infiltration and dermal hyperplasia leading to lesion formation [209]. The typical lesion develops from a macule with yellow, pink, or red coloration depending on the host, to a papule, to a vesicle that fills with fluid, and then neutrophils invade and die creating pus and transforming the vesicle into a pustule. This pustule eventually erupts into a raised hemorrhagic ulcerated lesion, the characteristic lesion of poxvirus infections [210]. This lesion usually forms a crust or scab that hardens, turns black, and falls off leaving a scar. Depending on the efficiency of host immune responses, this infection can disseminate and multiple, and possibly generalized lesions covering the host’s body can develop on the skin and mucous membranes [82]. In animals, especially cats, there are usually upper respiratory signs of disease such as nasal and ocular discharge [211]. Infections in animals non-native to Eurasia, such as zoo animals, can be often fatal with generalized lesions and additional symptoms including anorexia, lethargy, and dyspnea. In-depth reviews cover species-specific symptoms [82,212]. In humans, most immunocompetent patients present with a single painful lesion at the inoculation site, though additional symptoms such as fever (pyrexia), lethargy, sore throat, and general malaise are usually of enough concern to warrant time off and visits to health care professionals [211]. CPXV disease and symptoms usually resolve in 6–8 weeks [19], although cases of long-lasting generalized CPXV infection have been observed in patients with some form of immune-compromising conditions including eczema (atopic dermatitis) [142] and Darier’s disease [213]. Fatal cases of CPXV in humans have also been observed and are related to medical-induced immune suppression or HIV-induced immune dysregulation [152,214,215]. These cases are covered in more depth in the outbreaks section below.

## 9. Reservoir Hosts and Geographic Range

Determining the reservoir hosts of CPXVs, as well as the viruses’ geographic range, has been of great importance in understanding the etiology of outbreaks. Early attempts to find a reservoir host tested cattle serum for orthopoxvirus antibodies and found a prevalence of 0.7% in 1076 tested specimens [163]. Domestic cats were then deemed unlikely reservoirs due to symptomatic infections and evidence of interactions with rodents in their acquisition of infection [164]. As a result, serological surveys looking at trapped rodents’ serum antibodies’ ability to neutralize CPXV and immunofluorescent assays (IFA) began to be enacted. A major study conducted in England and Wales from 1975 to 1993 found low antibody prevalence in bank voles (*Myodes glareolus*), field voles (*Microtus agrestis*), wood mice (*Apodemus sylvaticus*), and one house mouse (*Mus musculus*) [216]. These species were tested for their susceptibility to viral infection with CPXV L97, showing no clinical signs of infection other than at inoculation sites at low PFUs. These species subsequently developed antibody responses post infection indicating some support for observed antibody seroprevalence and CPXV susceptibility [217]. A pivotal paper in defining CPXV geographic range and reservoirs was Cowpox: Reservoir Hosts and Geographic Range [218] published in 1999. Serology by IFA for CPXV antibodies was conducted with small, trapped rodents in woodland sites in England. PCR on the fusion gene of orthopoxviruses from a select number of specimens was used to generate a phylogeny supporting a CPXV designation at the time. The serology results supported the likely reservoirs of CPXV in Northwest England as bank voles and wood mice, though wood mice tested in Northern Ireland, where no cases of CPXV have been documented, were all seronegative. A subsequent paper tested the transmission dynamics of CPXV in bank voles and wood mice in the UK, finding that transmission was population frequency-dependent as opposed to population density-dependent and that transmission between the two species was negligible [219]. An analysis of host population dynamics in bank voles and wood mice found that young females infected with CPXVs had delayed maturation and delayed reproduction, usually until the next breeding season, indicating there are costs to CPXV infection even in potential reservoir hosts [220].

Chantrey et al. also analyzed confirmed CPXV cases to infer the geographic range of CPXVs, defining the range as “an area approximately bounded by Norway and Northern Russia, Moscow, Turkmenia, Northern Italy, France, and Great Britain” [218]. They also highlighted two species that could be considered reservoirs: wild rats (*Rattus norvegicus*) and the root or tundra vole (*Microtus oeconomus*), which had confirmed CPXV cases but live in much larger ranges beyond observed CPXV cases, with rats being found worldwide and root voles living throughout Russia and Alaska. Many other exploratory surveys for CPXV in trapped rodents throughout Europe found evidence of orthopoxvirus and/or CPXV infection but failed to directly isolate infectious virions [221,222,223,224,225,226,227]. Two of these types of surveys identified CPXV antibodies in rodents far outside the range of confirmed cases: One identified CPXV antibodies by IFA in wild rodents as far as the Baikal region in the Buryatia Russian Republic in 4/60 striped dwarf hamsters (*Cricetulus barabensis*), 7/107 reed voles (*Microtus fortis*), 1/34 tundra voles, and 2/103 gray red-backed voles (*Myodes rufocanus*) [228]. The second study identified potential CPXV by IFA and PCR fragment size on a gel in Turkey in a screen including arenavirus and hantavirus, although only in 1/330 wood mice was a positive sample found [229]. These findings might indicate that previously undetected CPXVs or other orthopoxviruses that generate cross-reactive antibodies/PCR fragments are present in the surveyed regions.

One major finding that eluded researchers was the isolation and genomic study of a CPXV from a suspected reservoir host. It was not until 2015 that the first isolate of CPXV from a putative reservoir host was obtained and completely sequenced, and phylogenetically characterized as discussed in the previous section [136]. The isolate was obtained from a common vole. CPXV infection was confirmed by electron microscopy, ATI formation, and later whole genome sequencing. High titer experimental infections with this isolate, named CPXV FM2292, in common voles and Wistar rats resulted in mild clinical symptoms including sneezing, nasal discharge, and shortness of breath, which disappeared quickly in the common vole group, whereas rats had symptoms for 16 days and developed poxvirus lesions. Low titer inocula resulted in no observed symptoms at all in common voles, supporting reservoir host type of infection, while Wistar rats had severe respiratory disease. Soon after, a CPXV isolate was identified from a rodent-trapping survey of a bank vole in Thuringia, Germany, by quantitative PCR (qPCR) and whole genome sequencing [160].

While surveys of CPXV in rodent populations make compelling claims of vole species and other rodents as reservoir hosts, especially bank voles, definitions of reservoir host can be contradictory and difficult to conclusively prove. One publication [207] argues that a reservoir be defined “as one or more epidemiologically connected populations or environments in which the pathogen can be permanently maintained and from which infection is transmitted to the defined target population,” where the target population is the population of interest, usually humans. More meta-level definitions such as Ashford’s—“a reservoir of infection is best defined as an ecological system in which the infectious agent survives indefinitely” [230]—argue that we should think of reservoirs as systems of interactions between organisms that propagate disease. The reservoir host situation in CPXVs case is not clear-cut. Viral diversity, combined with a large host and geographic range, makes the definition of reservoirs difficult. Modern techniques to survey, sequence, and monitor potential reservoirs will continue to make the situation clearer, though we may have to rethink CPXV maintenance in the wild as a larger system beyond single reservoirs as indicated by the works cited above.

## 10. Modern CPXV Hosts and Outbreaks

As alluded to and described in some detail in the above text, throughout the history of CPXV, various outbreaks have occurred throughout Eurasia in diverse mammalian species, starting with “cow-pox” outbreaks in farm cattle up until the mid-1970s [163,231] to a growing number of outbreaks in zoos beginning in the 1960s, and finally with the rising present-day cases in humans and domesticated animals, especially in cats. Below we present a brief overview of modern CPXV cases starting in the 1960s, highlighting in-depth reviews on the subject. We then analyze unique animal and human outbreaks in-depth with a focus on possible transmission events, diagnosis, and host outcomes.

While CPXV outbreaks were once described as extremely rare, increasing case studies and related publications have documented a substantial rise in CPXV infections with over 30 major outbreaks [18] and 54 reported human cases by 1994 [19]. However, the majority of human cases have occurred post 1994 and are covered below. Despite the name, modern CPXV isolates have rarely been derived from cattle, with the last known case from 1976 in Taunton, England, where cattle and farm workers were infected [163] indicating that cattle are incidental hosts of CPXVs, such as many of the various CPXV hosts [82]. Infections of these diverse hosts began to be observed in the 1960s with modern CPXV outbreaks and continue to the present day as many European zoos and farms have experienced devastating fatal CPXV infections of non-endemic exotic animals, such as elephants, rhinoceros, macaques, llamas, okapis, and multiple *Felidae* family members including lions, jaguars, and cheetahs, as well as other zoo-related hosts covered in great depth in [18,36,82,212,232]. Cases from other mammals including suspected rodent reservoirs discussed previously, cats, dogs [164], and horses are also well documented [232,233]. Because many of these hosts did not have CPXV isolates taken, preserved, or sequenced, especially from earlier outbreaks, the genetic classification of these CPXVs is unknown, though in most cases, location, PCR, and orthopoxvirus positive antibody tests support CPXV classifications. According to our estimates, using hosts from fully sequenced CPXVs and other confirmed cases, 60 unique species have been infected by CPXVs, although the list of potential susceptible species is likely much larger [82,212,218,227,229,232]. Our current knowledge of CPXV hosts from isolates that have had their whole genome sequenced is shown in Figure 3.

CPXV cases in zoos were historically most frequent in elephants with over 60 cases, leading to elephant vaccination against orthopoxvirus using MVA in Germany [36]. Conditions at zoos made viral transmission optimal, as animals were generally housed in facilities with multiple other animals in proximity and close contact with human zookeepers as vectors between them. It has also been theorized that in these environments, wild rodents including suspected CPXV reservoirs such as wood mice, voles, and rats may have been able to infiltrate and spread disease [234,235]. White rats (*Rattus norvegicus*) bred for food purposes in these zoos have also been implicated as potential vectors [18,85]. Additionally, the stress of zoo captivity may have had effects on the normal functioning of animal immune systems making them more susceptible to viral infection. This is possibly due to their unnatural environment and diet and/or changes to glucocorticoid (GC) levels, an area of research still under intense scrutiny with vast species-specific GC level differences between wild and captive animals [236].

The largest zoo outbreaks with cross-species transmission based on Kurth and Nitsche [82] include: the Moscow, Russia, outbreak of 1973–1974 [85]; the Berlin, Germany, outbreak in 1997 [237]; the Almere, Netherlands, outbreak in 2003 [235]; and the Krefeld, Germany, outbreak of 2008 [18], covered in depth in their respective citations. As an example, the Krefeld outbreak led to the death of two infected banded mongooses (*Mungos mungo*) while the other 13 mongooses were euthanized. One of two infected jaguarundi (*Herpailurus yagouaroundi)* also died of CPXV, which was confirmed by qPCR and infection of CAM resulted in hemorrhagic lesions. These outbreaks demonstrated the broad host range, transmissibility, and lethality of CPXVs, as many animals died or had to be euthanized due to their infections. Below, more animal-focused CPXV outbreaks are highlighted.

In the summer and fall of 2002, a massive outbreak of CPXV infected members of a colony of 80 new world primates located in Lower Saxony, Germany, including marmoset and tamarin species, killing 30 [148]. A diagnosis of CPXV was supported by electron microscopy of poxvirus virions, ELISA, and PCR of the D8L gene (CPXV125/OPG120) [161]. It was assumed that rodents transmitted CPXV to the primates after a flood as many rodents were observed around the cages. Trapping and testing revealed orthopoxvirus antibodies in 41% of the captured mice. Despite contact with humans, no human cases were reported.

Another lethal outbreak of CPXV occurred in four cotton-top tamarins (*Saguinus oedipus*) in Germany during September 2010 [158]. The tamarins and some nearby caged marmosets tested positive for CPXV by qPCR, although no endemic rodents tested positive leaving the potential vector or cause unknown.

Four outbreaks of CPXV throughout 2012–2017 in alpacas (*Lama pacos*) in four herds in Eastern Germany (in the states of Thuringia, Saxony-Anhalt, Saxony, and Brandenburg) led to generalized pocks, lesions, and fatalities [149]. Using indirect IFA for CPXV-specific antibodies, 28/107 alpacas tested positive. In addition, after testing of multiple species of local rodents, samples from bank voles and striped field mice tested positive for antibodies by IFA. A cat with clinical signs also tested positive for CPXV by PCR on the HA gene (CPXV194/OPG185) [161], and a CPXV isolate Ger/2013/Cat/Kira was isolated from the cat’s skin. Careful epidemiological investigations of alpaca pens identified one dead common vole, which was found drowned in a drinking water bucket for the alpacas and from which a CPXV was isolated and fully sequenced. The sequence of this isolate (Ger/2017/common vole FMEimka) was 99.997% identical to ones found in the alpacas providing rare evidence of transmission from the suspected reservoir to incidental hosts.

Episodes of recurring CPXV cases, in the summer and fall months, plagued a safari park in Denmark between 2010 and 2014 infecting nine cheetahs (*Acinonyx jubatus*) including two deaths [156]. CPXV diagnosis was first confirmed by real-time PCR specific for orthopoxviruses and CPXV, and whole genome sequencing [147]. Testing of other large cats resulted in seropositive test results in healthy lions and one tiger. Local environmental testing revealed that 14/21 local water voles (*Arvicola amphibius*) tested positive for orthopoxvirus antibodies, a presumed vector for this outbreak. Of note is the observed seasonality of outbreaks in the cheetahs with increased cases in late summer and fall months. This phenomenon is supported by correlations in seasonal increases in the proportions of positive antibody titers in wood mice and bank voles in the United Kingdom in late summer and fall, which correlates with the yearly maximum of rodent populations [151,218]. This seasonal shift in reservoirs has also been correlated with the increased incidence of CPXV in free-roaming domestic and stray cats [164,238,239] and humans [19] indicating a possible infection chain. Many CPXV infections have been detected in cats, with over 400 cases in western Eurasia up to 2004 alone [45]. It has been postulated that generally cats interact with infected rodents by hunting and ingesting infected rodents or are bitten by their infected prey [240]. A survey of German veterinarians provided an in-depth analysis of cat-related CPXV clinical cases with a focus on epidemiology, finding 17% of apparently unaffected cats seropositive for orthopoxviruses and revealed two cases of human CPXV cases likely acquired from cats [239]. Additional support for this infection chain comes from claims that approximately 50% of human zoonotic CPXV transmissions were attributed to cats with signs of rodent interactions such as infected bite wounds that developed into lesions on the head, neck, or forelimbs [241,242]. While most infected cats recover, an unusual recent outbreak of cat CPXV thought to be acquired nosocomially was more deadly killing three out of five infected cats [151]. All cases presented with lameness and dermal changes affecting the hindlimbs, symptoms not usually associated with cat CPXV infections. Although HA gene sequencing showed matches with other CPXV strains for four out of five isolates, whole genome sequencing that was later completed demonstrated that all isolates belong to the same clade. However, presently it is unknown if these viruses’ unique clinical presentation in cats is representative of a CPXV species uncharacterized properties, though other cases of deadly CPXV infections with unique presentations such as pneumonia have been reported [243]. Of note is the increasing number of human cases of CPXV infections since the cessation of smallpox vaccination after the eradication of naturally circulating VARV. Most human CPXV cases have clear connections to zoonotic exposure to infected animals, usually domesticated cats although dogs, farm animals, and more exotic zoo animals have transmitted CPXV to humans as well [142]. Below we cover the wide range of modern human CPXV cases including noteworthy and rare cases that emphasize transmission and health concerns with rising CPXV incidence.

A cat-transmitted CPXV infection led to the death of an unvaccinated 18-year-old man in 1990 who was immunosuppressed as part of his treatment for severe endogenous eczema and allergic asthma bronchiale [152]. The disease started with a few lesions that contained poxvirus virions as determined by electron microscopy, IFA, and trademark hemorrhagic pocks on CAM. Lesions developed into a “generalized, confluent pox exanthema with haemorrhage.” Despite drastic medical interventions, the patient succumbed to the disease, dying of a related pulmonary embolism. This first reported death from CPXV highlighted the risk to individuals with immune-related skin conditions and immunosuppression. Other documented cases of CPXV-related death include a 35-year-old man who was HIV positive and died of CPXV-related septic shock [214], a 17-year-old male under immunosuppression to reduce graft versus host disease post renal transplantation who may have acquired CPXV from a family cat and died of related multiorgan failure [215], and a similar case of lethal generalized CPXV in a immunosuppressed 32-year-old male kidney transplant patient in 2021 [165].

In eastern Finland, a 4-year-old girl with atopic dermatitis living on a farm acquired a generalized form of CPXV and was hospitalized in 2000. Infection by an orthopoxvirus was confirmed via electron microscopy, HA PCR, and sequencing, which confirmed a close genetic relationship to CPXVs. After whole genome sequencing, this isolate was named CPXV_FIN2000_MAN [134]. The patient recovered and additional investigation of orthopoxvirus responsive antibody prevalence in family pets showed high titers in their dogs, presenting one possible route of infection. Sampling in the area revealed positive orthopoxvirus antibody titers in cats, horses, and wild rodents.

In 2002 in the Netherlands, a 14-year-old girl taking care of a sick rat (*Rattus norvegicus*) acquired a CPXV infection confirmed by PCR based on the gene coding for the orthopoxvirus fusion protein identified in [244]. The rat died six days later and was tested positive for CPXV by PCR, which yielded identical results to the girl’s [245]. A similar 2008 outbreak of CPXV infections in white fancy pet rats and their owners around Krefeld, Germany, led to six human cases [144]. This marked the start of the 2008–2011 pet rat CPXV outbreak. By early 2009, neighboring countries including France [124] and the Netherlands reported six seemingly related cases, and 21 additional German cases [18,20,82]. A CPXV transmission from a pet rat to a woman in France related to the France–German outbreak in 2011 led to a severe ear chondritis, highlighting that even in immunocompetent individuals there can be drastic effects of CPXV infection [246]. Although a reliable source of this outbreak has not been identified, an investigation uncovered the wholesale supplier of the pet rats near the German–Czech border as the unique point in common for this large outbreak. A re-evaluation of this outbreak published in 2019 found that throughout 2008 and 2011, 40 human cases of CPXV infections acquired from pet rats were observed in France and Germany [21]. During this outbreak, the CPXV isolates maintained extreme sequence integrity as the sequenced whole genomes had a maximal difference of three single nucleotide variants over 4 years. One isolate of note from this outbreak was a CPXV from a lesion in a *Boa* genus snake that had been bitten by a rat purchased as food for the boa. If the virus was able to replicate in the boa and was not just recovered from initial inoculum transmitted from the bite, this would be the first non-mammalian host of a CPXV, and potentially of all orthopoxvirus members, to the authors’ knowledge and as presented in recent reviews that discuss orthopoxvirus host range [171,212]. Although boas might be potential hosts of CPXV, this requires more support since such a jump in host range between classes of animals (Mammalia to Reptilia) has yet to be supported/observed in orthopoxviruses. Of note, it has been suggested that snakes might be hosts for some orthopoxviruses based on the presence of a short interspersed element, putatively derived from a West African carpet viper (*Echis ocellatus*) or close relative in the tatera poxvirus genome, which was isolated from a healthy Kemp’s gerbil (*Tatera kempi*) [247].

In 2007, a unique CPXV transmission chain involving three species was identified from a rat to an Asian elephant (*Elephas maximus*) to a human [234]. After the unvaccinated elephant’s euthanasia and confirmation of CPXV by CAM red pocks, IFA, and PCR, a 19-year-old zookeeper, who also had not been vaccinated with a VACV vaccine, developed a lesion from which a CPXV HA gene was sequenced that was identical to that recovered from the elephant. Epidemiological investigations into the rodent population identified four infected rats with identical HA gene sequences but no clinical signs, indicative of a transmission chain between the three species.

A special case of CPXV occurred in 2016 in Amiens, France, where a 45-year-old male electrician was cut by a guardrail’s sharp end, usually stored in the ground, highlighting the potential transmission of CPXV via fomites [157]. The wound developed into a black eschar that did not respond to antibiotic treatment and the patient went to the hospital for treatment and diagnosis, where PCR confirmed an orthopoxvirus infection and whole genome sequencing confirmed a CPXV designation. Interestingly, the patient was vaccinated against smallpox with VACV as a one-year-old, 44 years prior to the incident, indicative of waning immunity supported by research findings of the best protection for within 10 years after vaccination [248], while other studies have claimed decades-long immunity [249]. The patient also owned an outdoor cat, although a veterinarian’s examination found no signs of CPXV, and the patient was certain he was not scratched by his cat; however, this route cannot be fully ruled out.

In 2017, a CPXV of unknown etiology infected a pregnant mother, leading to the death of her fetus [250]. Orthopoxvirus congenital infections in fetuses have been documented before in VARV infections [251] and as a rare complication of VACV infection leading to premature birth or death [252]. Preceding her fetus’s congenital infection with CPXV, the patient had atopic dermatitis localized to her hands and lived in a household near a farm where a cat, dogs, and rabbits roamed outside, although none of them were tested for orthopoxvirus infections. The patient claimed to have not touched her animals or cleaned their litter since the start of her pregnancy. CPXV infection was confirmed in fetal and placental samples by PCR of D8L (CPXV125/OPG120) [161] and D11L (CPXV128/OPG123) [161] with virions identified by electron microscopy. Twenty days after the first symptoms, the fetus was declared dead based on echography. This case once again highlights the danger of CPXV infection in immunocompetent individuals, especially pregnant mothers.

Despite the overall low incidence compared to other diseases, CPXV outbreaks are of concern with frequent seasonal outbreaks across mammalian species and increasing human cases. In addition, CPXV infections might be undiagnosed in cases with low morbidities and minor symptoms. In captive exotic animals, the risk of lethal infection is increased, with high fatality rates [82]. In humans, high risks for sequelae or death due to CPXV infection have been observed in immunosuppressed individuals due to medicinal or natural causes and those with inflammatory skin conditions, especially those with atopic dermatitis (eczema) as seen in generalized VACV infections [253]. Identified fatalities in various species, including the human cases presented here, demonstrate a potential for harm that must be monitored and prepared for by stocks of available VACV vaccines for at-risk patients, followed by animal monitoring and vaccination where appropriate. Improvements to captivity environments in zoos and farms that limit wild rodent interactions will help to mitigate the risk and incidence of CPXV outbreaks and other zoonotic diseases. Although no large human-to-human transmission chains have been observed, the potential has been proposed [254], and new transmission routes such as sexual contact as documented in the 2022 global mpox outbreak [255,256,257] could possibility spread CPXVs. Poxviruses have highly adaptable genomes and can tolerate major genomic changes, including gene duplications and deletions, recombination with closely or distantly related poxviruses, extension or contraction of the inverted terminal repeat regions, and the uptake of foreign genes through horizontal gene transfer [258]. The increased cases of CPXV in humans can augment the chance that genetic adaptions arise that contribute to better replication and transmission in humans and our close animal companions. Because CPXVs have the largest gene content among orthopoxviruses and the broadest host range, better adaption to humans is a major concern, which is emphasized by the discovery of VARV-related CPXVs. Continued monitoring and research into CPXVs and medical interventions against these viruses remain our best defense against the possibility of future large-scale outbreaks.

## 11. The Future of CPXVs

The name cow-pox has meant many different things at various points in history. In Jenner’s account, it was an immemorial disease of dairies known time immemorial. For a long time after the development of smallpox vaccination, the distinct identities of CPXV and VACV became obscure, sharing claimed origins from cow-derived vaccine strains. Only through the efforts and observations of Downie in the late 1930s, CPXV began to have a modern context as a distinct virus derived from a cow with phenotypic characteristics including A-type inclusions and red pock formation on CAM. Pox or pock-inducing infections throughout European zoos and farms were eventually subsumed into this classification due to their phenotypic characteristics and location, which only modern genetic analyses and phylogenies have revealed as multiple species constrained under the CPXV designation. These viruses are in need of reclassification, and while proposals exist that maintain the legacy of host-based naming, long-lasting nomenclature should attempt to be descriptive of the unique properties of these viruses without inciting bias based on location or historically inaccurate and narrow descriptions of hosts. The search for these properties in the genome and through experimental infections of animal models has distinguished some of these viruses but has not generated obvious properties on which to differentiate them, and perhaps future proposals to the ICTV will have to resort to name conventions somewhat removed from descriptive properties.

In the context of two major zoonotic outbreaks, the ongoing SARS-CoV-2 pandemic and the 2022 global mpox outbreak, public health concerns that another orthopoxvirus zoonosis could spread worldwide and cause human disease are not unfounded. CPXVs were ranked number 28 on a potential list of threatening animal-to-human zoonotic viruses [259], with a large host range and increasing human and animal cases. Although we have vaccines that elicit strong cross-protective immunity against orthopoxvirus infection [260,261] and antivirals such as cidofovir, (S)-HPMP-5-azaC, CMX001, and ST-246 specific for managing OPV-related infections, and these have been tested against different CPXVs in vitro [203], our collective ability to mobilize these therapeutics globally when needed has been criticized [262,263,264]. Thankfully today, CPXVs remain relatively benign and rare infections in immunocompetent humans; however, in animals, especially in those species non-native to Eurasia, CPXV infection can be often fatal. Continued surveillance of suspected reservoirs and seasonal adjustments, such as keeping domestic outdoor cats indoors during late summer and early fall months during peak rodent populations and CPXV infections, may reduce the transmission of CPXVs to humans. Improvements to the containment of these animals that minimize potential exposure to infected rodents may help lower zoo outbreaks, though the feasibility of such initiatives is low due to financial concerns. Despite these facts, we must continue to research CPXVs and prepare for possible outbreaks of larger proportions for the health of our animal companions and ourselves.

## Figures and Tables

**Figure 3 biomolecules-13-00325-f003:**
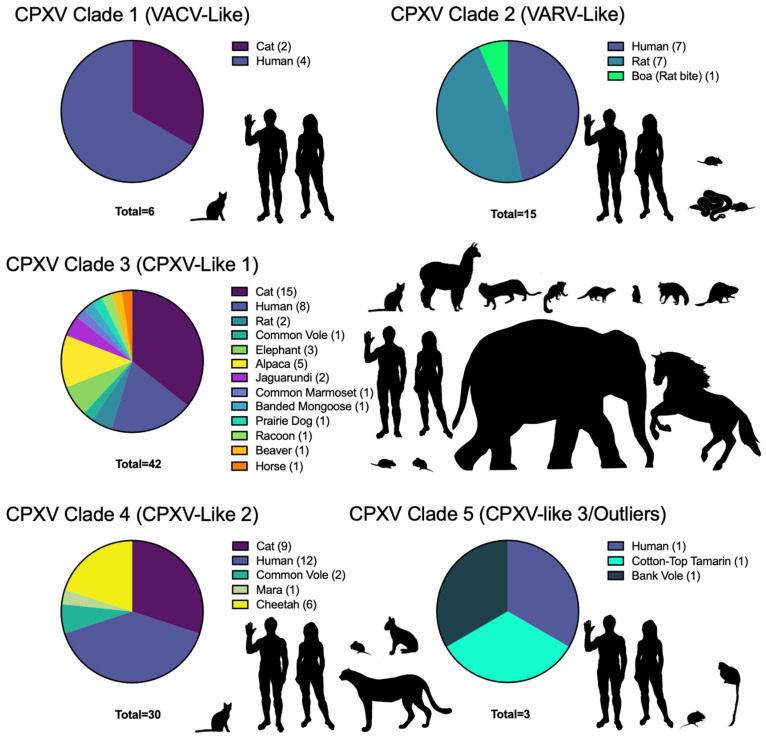
Hosts of fully sequenced CPXV isolates. Pie charts represent isolates in CPXV major clade classifications based on Figure 1 (Franke et al. [14] designations in parentheses) and the proportion of the pie represents the number of isolates from that host species. Legend to the right of each chart has the common names of host species, number of cases in parentheses, and follows the clockwise orientation of chart slices. Silhouettes are representative of the infected species. For image rights and licenses see Appendix A.

**Table 1 biomolecules-13-00325-t001:** Currently Available Fully Sequenced Genomes from Cowpox Viruses.

CPXV Classification	Name	Location	Year	Host	Accession Number	References
CPXV Clade 1	CPXV_Catpox3L97	UK	1977	Cat	KY549143.2	[126]
CPXV Clade 1	GRI-90	Russia, Moscow	1990	Human	X94355.2	[141]
CPXV Clade 1	Austria 1999	Austria, Texing	1999	Cat	HQ407377.1	[134]
CPXV Clade 1	Finland_2000_MAN	Finland, Tohmajärvi	2000	Human	HQ420893.1	[134,142]
CPXV Clade 1	HumLit08/1	Lithuania, Vilnius	2008	Human	KC813493.1	[135]
CPXV Clade 1	Kostroma_2015	Russia, Kostroma	2015	Human	KY369926.1	[143]
CPXV Clade 2	HumAac09/1	Germany, Aachen	1990	Human	KC813508.1	[135,144]
CPXV Clade 2	HumGra07/1	Austria, Graz	2007	Human	KC813510.1	[135,144]
CPXV Clade 2	HumKre08/1	Germany, Krefeld	2008	Human	KC813512.1	[135,144]
CPXV Clade 2	RatKre08/2	Germany, Krefeld	2008	Rat	KC813505.1	[135,144]
CPXV Clade 2	Ratpox09	Germany, Munich	2009	Rat	LN864565.1	[136,145]
CPXV Clade 2	RatGer09/1	Germany, Germering	2009	Rat	KC813503.1	[135,144]
CPXV Clade 2	RatAac09/1	Germany, Aachen	2009	Rat	KC813501.1	[135,144]
CPXV Clade 2	CPXV/Rat Koelle	Germany, Munich	2009	Rat	MK035757.1	[21]
CPXV Clade 2	CPXV/Compiegne 2009	France, Compiègne	2009	Human	MK035750.1	[21]
CPXV Clade 2	CPXV/Rat Biederstein	Germany, Munich	2009	Rat	MK035749.1	[21]
CPXV Clade 2	CPXV/Boa Marl	Germany, Marl	2009	Boa	MK035748.1	[21]
CPXV Clade 2	CPXV/Rat Marl	Germany, Marl	2009	Rat	MK035747.1	[21]
CPXV Clade 2	CPXV/Boy Biederstein	Germany, Munich	2009	Human	MK035746.1	[21]
CPXV Clade 2	CPXV/Cepad 332	France, Épinal	2011	Human	MK035759.1	[21,146]
CPXV Clade 2	CPXV/Cepad 333	France, Épinal	2011	Human	MK035753.1	[21,146]
CPXV Clade 3	Germany_1971_EP1	Germany	1971	Elephant	KY463519.1	[147]
CPXV Clade 3	Germany_1980_EP4	Germany, Hameln	1980	Elephant	HQ420895.1	[87,134]
CPXV Clade 3	CPXV_K780	UK	2000	Cat	KY549147.1	[126]
CPXV Clade 3	CPXV_K779	UK, Bristol	2000	Cat	KY549146.1	[126]
CPXV Clade 3	CPXV/75/01	Germany, Detmold	2001	Human	MK035755.1	[21]
CPXV Clade 3	Germany_2002_MKY	Germany, Göttingen	2002	Common Marmoset	HQ420898.1	[134,148]
CPXV Clade 3	BeaBer04/1	Germany, Berlin	2004	Beaver	KC813491.1	[135]
CPXV Clade 3	CPXV/S2216/04	Germany, Hannover	2005	Cat	MK035751.1	[21]
CPXV Clade 3	HumGri07/1	Germany, Grimmem	2007	Human	KC813511.1	[135]
CPXV Clade 3	HumBer07/1	Germany, Berlin	2007	Human	KC813509.1	[135]
CPXV Clade 3	EleGri07/1	Germany, Grimmem	2007	Elephant	KC813507.1	[135]
CPXV Clade 3	CatPot07/1	Germany, Potsdam	2007	Cat	KC813506.1	[135]
CPXV Clade 3	CatBer07/1	Germany, Berlin	2007	Cat	KC813502.1	[135]
CPXV Clade 3	HumMag07/1	Germany, Magdeburg	2007	Human	KC813495.1	[135]
CPXV Clade 3	HumPad07/1	Germany, Paderborn	2007	Human	KC813496.1	[135]
CPXV Clade 3	MonKre08/4	Germany, Krefeld	2008	Banded Mongoose	KC813500.1	[18,135]
CPXV Clade 3	JagKre08/2	Germany, Krefeld	2008	Jaguarundi	KC813498.1	[18,135]
CPXV Clade 3	JagKre08/1	Germany, Krefeld	2008	Jaguarundi	KC813497.1	[18,135]
CPXV Clade 3	HumLan08/1	Germany, Landau	2008	Human	KC813492.1	[135]
CPXV Clade 3	RatHei09/1	Germany, Heidelberg	2009	Rat	KC813504.1	[135]
CPXV Clade 3	Ger/2010/Raccoon	Germany, Ellrich	2010	Raccoon	LT896730.1	[14]
CPXV Clade 3	Ger/2010/Cat	Germany, Nordhausen	2010	Cat	LT896729.1	[14]
CPXV Clade 3	Ger/2010/Alpaca	Germany, Oberwiesenthal	2010	Alpaca	LT896718.1	[14]
CPXV Clade 3	Ger/2010/Rat	Germany, Hannover	2010	Rat	LT896728.1	[14]
CPXV Clade 3	Ger/2012/Alpaca	Germany, Rositz	2012	Alpaca	LT896726.1	[14,149]
CPXV Clade 3	Ger/2013/Alpaca	Germany, Zernitz	2013	Alpaca	LT896719.1	[14,149]
CPXV Clade 3	Ger/2014/Cat1	Germany, Bleckede	2014	Cat	LT896723.1	[14]
CPXV Clade 3	Ger/2014/Cat2	Germany, Nordhausen	2014	Cat	LT896725.1	[14]
CPXV Clade 3	Ger/2015/Cat2	Germany, Rostock	2015	Cat	LT896727.1	[14]
CPXV Clade 3	CPXV Amadeus 2015	Germany, Berlin	2015	Horse	LN879483.1	[150]
CPXV Clade 3	Ger/2015/Human2	Germany, Leipzig	2015	Human	LT993232.1	[14]
CPXV Clade 3	Ger/2015/Prairie-dog	Germany, Dresden	2015	Prairie Dog	LT993231.1	[14]
CPXV Clade 3	Ger/2015/Cat3	Germany, Vogtlandkreis	2015	Cat	LT896733.1	[14]
CPXV Clade 3	Ger/2015/Cat4	Germany, Hengelbach	2015	Cat	LT896731.1	[14]
CPXV Clade 3	Ger/2015/Human1	Germany, Leipzig	2015	Human	LT896720.1	[14]
CPXV Clade 3	CPXV/Leo 2015/5	Germany, Extertal	2015	Cat	MK035756.1	[21]
CPXV Clade 3	CPXV/Cookie S963_15	Germany, Benthe	2015	Cat	MK035754.1	[21]
CPXV Clade 3	CPXV/Sammy 2015/4a	Germany, Bad Münder	2015	Cat	MK035752.1	[21]
CPXV Clade 3	CPXV/Moritz 2015/3a	Germany, Aerzen	2015	Cat	MK035758.1	[21,151]
CPXV Clade 3	Ger/2017/Alpaca2	Germany, Merzdorf	2017	Alpaca	LT896732.2	[14,149]
CPXV Clade 3	Ger/2017/Alpaca1	Germany, Brand-Erbisdorf	2017	Alpaca	LT993230.1	[14,149]
CPXV Clade 3	Ger/2017/common vole FMEimka	Germany, Brand-Erbisdorf	2017	Common Vole	LT993228.1	[149]
CPXV Clade 4	Brighton Red	UK, Brighton	1937	Human	AF482758.2	[17,70,77]
CPXV Clade 4	CPXV_Catpox5wv1	UK, London	1972	Cheetah	KY549144.1	[126]
CPXV Clade 4	No-Swe-H1	Sweden	1990	Human	OP125538.1	[118,140]
CPXV Clade 4	Germany_1990_2	Germany, Bonn	1990	Human	HQ420896.1	[152]
CPXV Clade 4	Germany 91-3	Germany, Munich	1991	Human	DQ437593.1	[117,153]
CPXV Clade 4	Norway_1994_MAN	Norway, Bergen	1994	Human	HQ420899.1	[134,154]
CPXV Clade 4	UK2000_K2984	UK, Bristol	2000	Cat	HQ420900.1	[134]
CPXV Clade 4	France_2001_Nancy	France, Nancy	2001	Human	HQ420894.1	[134]
CPXV Clade 4	Ger/2007/Vole	Germany, Rottweil	2007	Common Vole	LT896722.1	[14]
CPXV Clade 4	MarLei07/1	Germany, Leipzig	2007	Mara	KC813499.1	[135]
CPXV Clade 4	HumLue09/1	Germany, Lübeck	2009	Human	KC813494.1	[135]
CPXV Clade 4	FM2292	Germany, Rutesheim	2011	Common Vole	LN864566.1	[136]
CPXV Clade 4	Ger/2014/Human	Germany, Freiburg	2014	Human	LT993226.1	[14]
CPXV Clade 4	Ger/2015/Cat1	Germany, Vogtlandkreis	2015	Cat	LT896724.1	[14]
CPXV Clade 4	No-Swe-H2	Sweden	1990	Human	OP125537.1	[118,140]
CPXV Clade 4	Norwayfeline	Norway, Bergen	1994	Cat	KY549151.1	[126]
CPXV Clade 4	No-H1	Norway	1994	Human	OP125539.1	[118,140]
CPXV Clade 4	No-F1	Norway	1994	Cat	OP125541.1	[118,140]
CPXV Clade 4	Germany_1998_2	Germany, Eckental	1998	Human	HQ420897.1	[134,155]
CPXV Clade 4	No-F2	Norway	1999	Cat	OP125540.1	[140]
CPXV Clade 4	CPXV_K2739	UK, Bristol	2000	Cat	KY549149.1	[126]
CPXV Clade 4	CPXV_K4207	UK, Bristol	2000	Cat	KY549150.2	[126]
CPXV Clade 4	CPXV_1639	UK, Bristol	2000	Cat	KY549148.1	[126]
CPXV Clade 4	CPXV_K428	UK, Bristol	2000	Cat	KY549145.1	[126]
CPXV Clade 4	CPXV CheGrey_DK_2010	Denmark, Djursland	2010	Cheetah	KY569021.1	[156]
CPXV Clade 4	CPXV CheTopCut_DK_2011	Denmark, Djursland	2011	Cheetah	KY569022.1	[156]
CPXV Clade 4	CPXV CheNuru_DK_2012	Denmark, Djursland	2012	Cheetah	KY569020.1	[156]
CPXV Clade 4	CPXV CheHurley_DK_2012	Denmark, Djursland	2012	Cheetah	KY569018.1	[156]
CPXV Clade 4	CPXV CheNova_DK_2014	Denmark, Djursland	2014	Cheetah	KY569019.1	[156]
CPXV Clade 4	France Amiens 2016	France, Amiens	2016	Human	LT883663.1	[157]
CPXV Clade 5	Ger 2010 MKY	Germany, Bad Liebenstein	2010	Cotton-top Tamarin	LT896721.1	[14,158]
CPXV Clade 5	No-H2	Norway, Nordland	2001	Human	OM460002.1	[159]
CPXV Clade 5	GerMygEK 938/17	Germany, Thuringia	2017	Bank Vole	LR812035.1	[160]

CPXV classification based on clades observations in Figure 2.

## Data Availability

Data are included in this manuscript and in Appendix A.

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
