# Peer review of "Cowpox Viruses: A Zoo Full of Viral Diversity and Lurking Threats"

_biomolecules, 2023, doi:10.3390/biom13020325_

Round 1
Reviewer 1 Report
Cowpox Viruses: A Zoo Full of Viral Diversity and Lurking 2 Threats is a very nice review of the history of cowpox virus(es). It was well written and comprehensive. I have a few small suggestions to consider.
Double check the use of capitalization and italics for all references to viruses (see https://ictv.global/faq/names).
Change "monkeypox virus" to "Mpox virus" or "MPXV" throughout the paper.
Line 36. Add "an" before "orthopoxvirus that played a major role...".
Line 380. remove "a technique...DNA primers" as this just describes PCR itself, not RE digest of PCR fragments.
Line 392. Change "earlier" to "early".
Figure 2 is mentioned before Figure 1 in the text (Lines 576 & 603, respectfully). These figures should be reordered.
Line 1154. Reword the sentence "...it was a disease of dairies known time immemorial". Suggested change "...it was a immemorial disease of dairies".
The discussions of virus classification in sections 1 through 4 of the review are quite dramatic. I recommend toning down the wording in these sections just slightly to try and make the review sound more objective. Maybe point out the fact that our scientific knowledge and capabilities to classify viruses have improved by leaps and bounds only relatively recently. And convey some respect for the scientists whose work was robust given the limited tools they had at previous times in history. Also, if you can, please add a sentence or two discussing possible valid reasons why the ICTV has not changed the classification of the genetic subsets of viruses that are classified together as CPXV.
Expand section 5 (CPXV Genes and Host Interactions) slightly to improve the transition from the discussion of genetic differences in CPVX isolates to genes involved in host interactions. Additional references could be included in this section.
The transition from genetic differences to section 6 (Animal Models) was done very well. Adding a sentence or two about why there is a disconnect between the proposed genetic classifications and the mortality data in mouse models would be beneficial and interesting. Also, include at least one other reason that IL-6 is increased with increased mortality (e.g. more severe tissue damage leads to release of IL-6, rather than IL-6 promotes disease = essentially the chicken or the egg senario).
Figure 3 is really nice! Just one silly observation, you depict both female and male humans but all the other species only have one depiction. I guess they aren't dimorphic enough to add two of them to every picture? :)
Thank you for this review. I enjoyed reading it.
Reviewer 2 Report
Bruneau and co-workers provide an extraordinarily comprehensive review of cowpox viruses that will serve as a valuable reference. The endeavor to provide a complete scientific history will attract specialists, but may put-off non-specialists who would appreciate the first few sections on the early historical record but find the historical approach too detailed for some later sections. Aside from that, the review is accurate and clearly and logically written, and the selection of data and references unbiased. Congratulations to the authors for such a detailed review.
Minor Comments
Units 1 – 3. Good review of known early history of cowpox and vaccinia virus. Although not directly related to cowpox might add a few sentences regarding recent cases of vaccinia virus infection in animals and humans in Brazil.
Unit 3 – repeated unit name – either should be unit 4 or merged with previous unit. Might add primary reference Smadel, J. E. and C. L. Hoagland (1942). "Elementary bodies of vaccinia." Bacteriol. Rev. 6: 79-110 instead or in addition to ref. 21
Unit 4 – Might be useful to mention that the overall structure of the vaccinia virus genome was determined in 1982 [Baroudy, B. M., et al. (1982). "Incompletely base-paired flip-flop terminal loops link the two DNA strands of the vaccinia virus genome into one uninterrupted polynucleotide chain." Cell 28: 315-324.] and that the complete genome of vaccinia virus was sequenced in 1990. [Goebel, S. J., et al. (1990). "The complete DNA sequence of vaccinia virus." Virology 179: 247-266; 517-563.]
Line 597 – the term OPG is introduced without explanation. It would help to have a sentence that said something like: difficulty in comparing homologous genes among cowpox virus and other orthopoxviruses led to a uniform genus-wide nomenclature of othopoxvirus gene (OPG) names.
Unit 5 – Additional cowpox genes including B22 family proteins on T cell activation
Dasgupta, A., et al. (2007). "Cowpox virus evades CTL recognition and inhibits the intracellular transport of MHC class I molecules." J. Immunol. 178(3): 1654-1661.
Also, the interference with MHC-mediated antigen presentation by CPXV 012 and 203
Alzhanova, D., et al. (2009). "Cowpox virus inhibits the transporter associated with antigen processing to evade T cell recognition." Cell Host & Microbe 6(5): 433-445.
Luteijn, R. D., et al. (2014). "Cowpox Virus Protein CPXV012 Eludes CTLs by Blocking ATP Binding to TAP." Journal of Immunology 193(4): 1578-1589.
